# Reconstructing human pancreatic differentiation by mapping specific cell populations during development

Cyrille Ramond[1,2,3], Nicolas Glaser[1,2,3], Claire Berthault[4], Jacqueline Ameri[5], Jeannette Schlichting Kirkegaard[6], Mattias Hansson[7], Christian Honoré[6], Henrik Semb[5], Raphaël Scharfmann[1,2,3]*

[1]INSERM U1016, Cochin Institute, Paris, France; [2]CNRS UMR 8104, Paris, France; [3]University of Paris Descartes, Paris, France; [4]Immunology Department, Unit for Lymphopoiesisis, Paris, France; [5]The Danish Stem Cell Center (DanStem), Faculty of Health Sciences, University of Copenhagen, Denmark, Europe; [6]Department of Islet and Stem Cell Biology, Novo Nordisk A/S, Denmark, Europe; [7]Global Research External Affairs, Novo Nordisk A/S, Denmark, Europe

**Abstract** Information remains scarce on human development compared to animal models. Here, we reconstructed human fetal pancreatic differentiation using cell surface markers. We demonstrate that at 7weeks of development, the glycoprotein 2 (GP2) marks a multipotent cell population that will differentiate into the acinar, ductal or endocrine lineages. Development towards the acinar lineage is paralleled by an increase in GP2 expression. Conversely, a subset of the GP2+ population undergoes endocrine differentiation by down-regulating GP2 and CD142 and turning on *NEUROG3*, a marker of endocrine differentiation. Endocrine maturation progresses by up-regulating SUSD2 and lowering ECAD levels. Finally, in vitro differentiation of pancreatic endocrine cells derived from human pluripotent stem cells mimics key in vivo events. Our work paves the way to extend our understanding of the origin of mature human pancreatic cell types and how such lineage decisions are regulated.

*For correspondence: raphael.
scharfmann@inserm.fr

**Competing interests:** The authors declare that no competing interests exist.

## Introduction

Intensive efforts are currently dedicated towards the development of cell replacement therapies using cell types derived from human pluripotent stem cells (hPSC). Human insulin-producing beta cells represent a paradigm for this type of objective. These cells have a major physiological function, regulating circulating glucose levels by producing and secreting insulin. In patients suffering from type one diabetes, these cells are destroyed by an autoimmune mechanism, and would thus need to be replaced (*Benthuysen et al., 2016*). Beta cell replacement holds immense promises for diabetic patients and current strategies have reached major milestones (*Pagliuca et al., 2014*; *Russ et al., 2015*; *Rezania et al., 2014*). However, it is well accepted that a more detailed understanding of beta cell development in human is required to generate unlimited functional human beta cells (*Johnson, 2016*).

The adult pancreas is composed of acinar cells that excrete enzymes into the duodenum through a ductal tree, and of endocrine cells (approximately 1% of the total pancreatic cells) that are clustered together forming the islets of Langerhans. The endocrine cells secrete hormones such as insulin (beta cells), glucagon (alpha cells) somatostatin (delta cells), pancreatic polypeptide (gamma cells) and ghrelin (epsilon cells). The pancreas develops from the primitive gut tube that evaginates into a dorsal and a ventral anlage (*Pan and Wright, 2011*; *Jennings et al., 2015*). Multipotent epithelial

pancreatic progenitors co-expressing the transcription factors PDX1 and NKX6-1 (*Nelson et al., 2007*; *Cebola et al., 2015*) proliferate upon signals (such as FGF10) from the adjacent mesenchyme (*Bhushan et al., 2001*) and subsequently differentiate into the acinar, ductal and endocrine lineages. Endocrine commitment is initially marked by the expression of a basic helix-loop-helix transcription factor, NEUROG3, and followed by the expression of the mature endocrine markers (*Gradwohl et al., 2000*; *Gu et al., 2002*).

Due to the difficulties associated with procuring staged human fetal tissues and the limited tools for their analysis, few data on human development is available and the majority of knowledge on tissue development derives from animal models. This also applies to the pancreas (*Jennings et al., 2015*) where only a limited number of studies have been performed on human pancreatic development. Theses studies demonstrate similarities but also differences between rodent and human pancreatic development (*Scharfmann et al., 2013*; *Jennings et al., 2015*; *Nair and Hebrok, 2015*).

Knowledge on human pancreatic development remains limited. More information exists concerning human hematopoietic cell differentiation thanks to the characterization and use of cell surface antigens that enabled to identify, quantify and purify hematopoietic stem cells and progenitors at different stages of their development (*Eaves, 2015*). By mirroring the hematopoietic field, we developed here an approach where cell surface markers are used to recapitulate the hierarchical sequence of human pancreatic development. Specifically, we characterized the expression levels of specific markers at different stages of human pancreatic development corresponding to 7 to 12 weeks of development (WD). First we purified human pancreatic epithelial cells by selecting cells positive for the transmembrane glycoprotein EPCAM and by excluding CD45$^+$ hematopoietic and CD31$^+$ endothelial cells. Next we segregated pancreatic epithelial cells into four populations based on the GP2 and CDH1 (ECAD) expression levels. We observed that the expression levels of GP2 and ECAD correlate with acinar, ductal and endocrine functions. By using the additional cell surface markers CD142 and SUSD2 we further refined endocrine cell differentiation. Finally, our development model also applies to the in vitro differentiation of hPSCs into pancreatic endocrine cells.

Taken together our work provides a novel approach to study human fetal pancreas development and bridges the path between in vivo and in vitro differentiation of human pancreatic endocrine cells.

## Results

### EPCAM expression is restricted to the epithelium in the human fetal pancreas

We tested if EPCAM can be used to purify human fetal pancreatic epithelial fraction enriched in pancreatic progenitors. CD31 and CD45 were used to exclude endothelial and hematopoietic cells respectively. Using this combination of antibodies on human fetal pancreatic cells (9.7WD), we detected three distinct fractions: the CD45$^+$/CD31$^+$ fraction (the endothelial/hematopoietic cells), the CD45$^-$CD31$^-$EPCAM$^-$ fraction, and the CD45$^-$CD31$^-$EPCAM$^+$ fraction (*Figure 1A*). To unveil which fraction contained the pancreatic progenitors, we assayed the expression of PDX1 and NKX6-1. Immunohistochemistry analysis of human fetal pancreatic sections showed that EPCAM$^+$ cells expressed PDX1 (*Figure 1B*). FACS analysis demonstrated that PDX1 and NKX6-1 were co-expressed in the CD45$^-$CD31$^-$EPCAM$^+$ fraction (*Figure 1C–E*). RT-qPCR analysis on sorted fractions confirmed that *PDX1* and *NKX6-1* expressions were restricted to the CD45$^-$CD31$^-$EPCAM$^+$ fraction (*Figure 1F*) whereas the CD45$^-$CD31$^-$EPCAM$^-$ fraction did not express pancreatic markers and most likely represent the mesenchymal pancreatic fraction (later referred as population M) (*Figure 1A*). These results suggest that a combination of the cell surface markers EPCAM, CD45, and CD31 can be used to purify the human fetal pancreatic epithelial fraction.

### GP2 and ECAD define four populations in the human fetal pancreatic epithelium that develop sequentially

Recently, GP2 was identified as a novel cell surface marker of the immature pancreatic progenitor cells derived from hPSC (*Ameri et al., 2017*). Furthermore, additional data indicate that ECAD levels are tightly modulated during endocrine differentiation (*Gouzi et al., 2011*). Therefore, we analyzed GP2 and ECAD expression levels in the CD45$^-$CD31$^-$EPCAM$^+$ fraction at 9.4WD. GP2 expression in

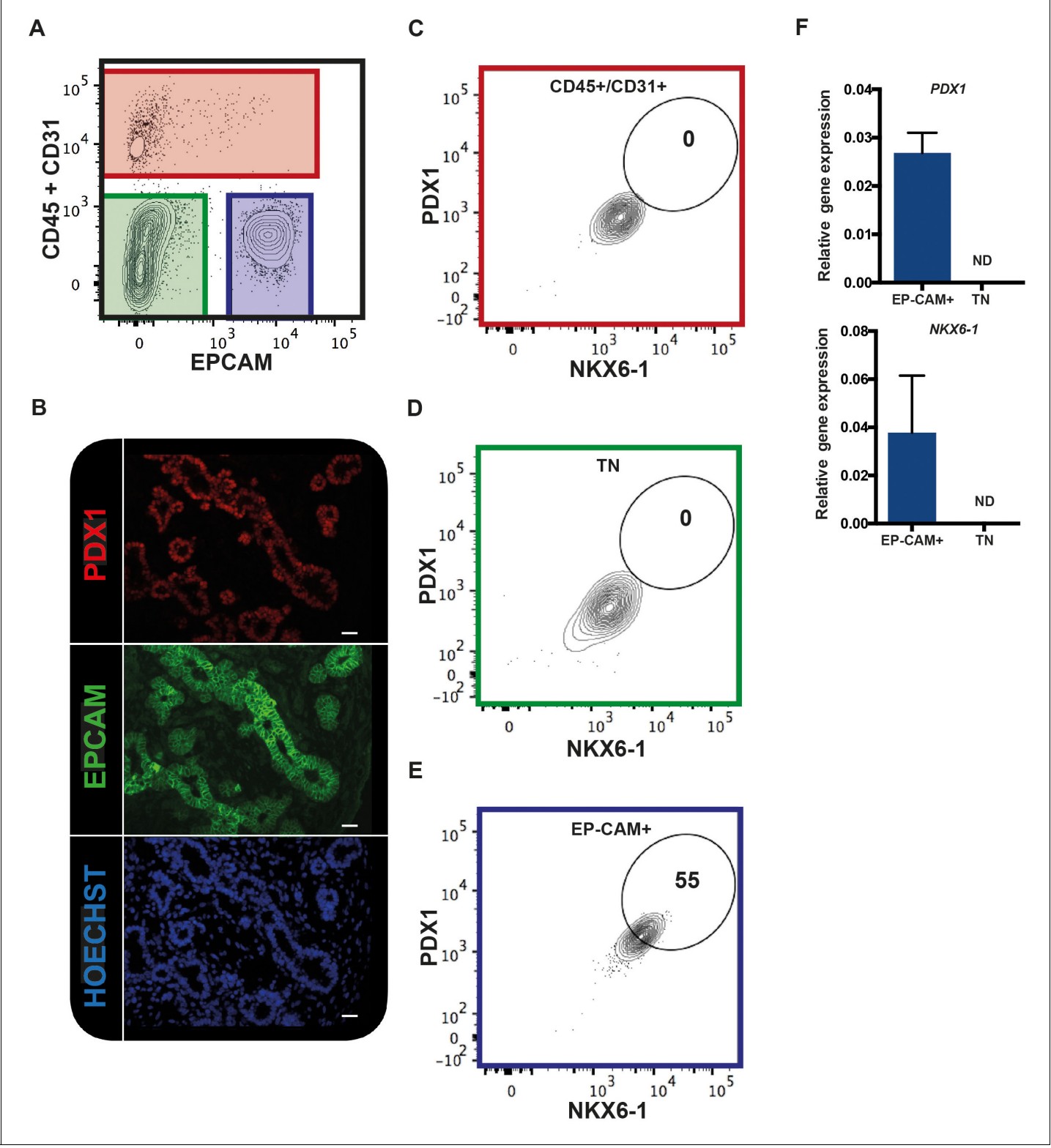

**Figure 1.** EPCAM expression in the human fetal pancreas. (**A**) The flow cytometry plot represents CD45 and CD31 expression against EPCAM gated on live human fetal pancreatic cells (9.7WD), n = 9. (**B**) Immunohistochemistry for PDX1 and EPCAM on pancreatic section (9WD), n = 3. Scale bar = 100 µm. (**C–E**) Flow cytometry plots of PDX1 and NKX6-1 expression at 9.4WD on CD45+/CD31+ cells (red square), CD45-CD31-EPCAM- cells (TN = triple negative green square) and CD45-CD31-EPCAM+ cells (blue square). (**F**) RT-qPCR analysis of *PDX1* and *NKX6-1* expression on sorted CD45-CD31-EPCAM+ and TN cells. ND = Not Detected.

CD45⁻CD31⁻ was restricted to the EPCAM⁺ fraction (*Figure 2—figure supplement 1*). GP2 and ECAD expression segregated the CD45⁻CD31⁻EPCAM⁺ fraction into four distinct populations: GP2$^{hi}$ECAD⁺ (named GP2$^{hi}$), GP2⁺ECAD⁺ (named GP2⁺), GP2⁻ECAD⁺ (named GP2⁻), and the GP2⁻E-CAD$^{low}$ (named E$^{low}$) (*Figure 2A*). Remarkably, the cell frequencies of the four populations were well conserved from one pancreas to the other, demonstrating inter-individual homogeneity (*Figure 2B*). We assayed the dynamic expression of GP2 and ECAD in the pancreatic epithelial fraction (CD45⁻CD31⁻EPCAM⁺) during development (from 7WD to 12WD) (*Figure 2C*). At 7WD the epithelial fraction was essentially GP2⁺ (96 ± 1%). From 7WD to 8.4WD, the GP2⁻ population evolves from 2 ± 1% to 34 ± 6% (p<0.05) (*Figure 2C,D*). Interestingly, from 8.6WD, we detected cells with a lower ECAD level (*Figure 2—figure supplement 2A,B*). This E$^{low}$ population further increased in frequency from 10 ± 2% at 9.4WD to 18 ± 3% at 12WD (p<0.05) (*Figure 2C,D*). At 9.4WD, we also detected a GP2$^{hi}$ population, increasing in frequency during development (3 ± 1% at 9.4WD to 20 ± 5% at 12WD; p<0.05) (*Figure 2C,D* and *Figure 2—figure supplement 2C*). Altogether, the temporal and continuum GP2 and ECAD expression suggest a progeny relationship between specific cell populations, the GP2⁺ population would differentiate either into a GP2$^{hi}$ population or into a GP2⁻ population that would later decrease its ECAD level to give rise to the E$^{low}$ population (*Figure 2E*).

## Acinar and endocrine functions segregate within the GP2 and ECAD populations

To characterize the four epithelial populations described above, we sorted the GP2$^{hi}$, GP2⁺, GP2⁻ and E$^{low}$ populations and performed global transcriptomic analyses combined with RT-qPCR analyses at 9 and 11WD. As a non-epithelial control, we included the CD45⁻CD31⁻EPCAM⁻ fraction (population M) (*Figure 1A*). Due to limitations in cell numbers, the GP2$^{hi}$ population was only sorted at 11WD. Principal component analysis (PCA) on the sorted populations at 9WD revealed three clusters. PC1 separated the epithelial populations (GP2$^{hi}$, GP2⁺, GP2⁻ and E$^{low}$) from the mesenchymal fraction (M), while PC2 segregated the E$^{low}$ population from the GP2$^{hi}$, GP2⁺ and GP2⁻ populations (*Figure 3A*). Gene Set Enrichment Analysis (GSEA) using Gene Ontology database indicated that digestion was the most represented biological process in the GP2$^{hi}$ and GP2⁺ populations while endocrine functions (insulin and peptide secretion, hormone secretion) were enriched in the E$^{low}$ population (*Figure 3B*, *Figure 3—figure supplement 2*). Conversely the pancreatic functions appeared less enriched in the GP2⁻ population. Next, we defined the list of the 'specific enriched genes' per population (*Supplementary file 1b-–c* and Materials and method section). We compared the fetal 'specific enriched genes' with the RNAseq Single Cell data from human adult pancreas (*Segerstolpe et al., 2016*) (*Figure 3—figure supplement 3*). The GP2$^{hi}$ population displayed 27 enriched genes (p<0.05), 85% being preferentially expressed in the adult acinar cells (*Figure 3C*, and *Figure 3—figure supplement 3A*). In contrast, the E$^{low}$ population contained 91 (at 9WD) and 34 (at 11WD) differentially expressed genes (p<0.05) that were also enriched (98% and 100% respectively) in the adult endocrine cells (alpha, beta, delta, epsilon or gamma cells) (*Figure 3C*, and *Figure 3—figure supplement 3B*).

We next generated heatmaps based on Gene Ontology lists and selected acinar, ductal and endocrine genes. By RT-qPCR analyses we confirmed that acinar markers such as *CEL*, *CELA3A* and *CTRC* were enriched in the GP2⁺ population at 9WD and in the GP2$^{hi}$ population at 11WD (*Figure 3—figure supplement 2A*, *Figure 4A*). Ductal markers (*CFTR*, *KRT19* and *SPP1*) were enriched in the GP2⁺ and GP2⁻ populations, especially *CFTR* at 9WD and 11-13WD (*Figure 4—figure supplement 1*). The E$^{low}$ populations (9 and 11WD) were enriched with endocrine transcription factors such as *ARX*, *PAX4*, *PAX6*, *NEUROD1*, *MAFA*, *MAFB*, *ISL1* and *NKX2.2* and hormones like *CHGA*, *GCG*, *GHRL* and *INS* (*Figure 3—figure supplement 2B*, *Figure 4B*).

Finally, to define the cell population where endocrine cells first differentiate, we followed the expression of the endocrine progenitor marker *NEUROG3*. *NEUROG3* was first detected at 8.4WD in the GP2⁻ population (*Figure 4C*) prior to the detection of the E$^{low}$ population (*Figure 2C*) and next (9 to 13WD), found enriched in the E$^{low}$ population (*Figure 4C*). Thus, our data suggest that the first human pancreatic endocrine progenitors differentiate in the GP2⁻ population and mature while decreasing ECAD levels in the E$^{low}$ population.

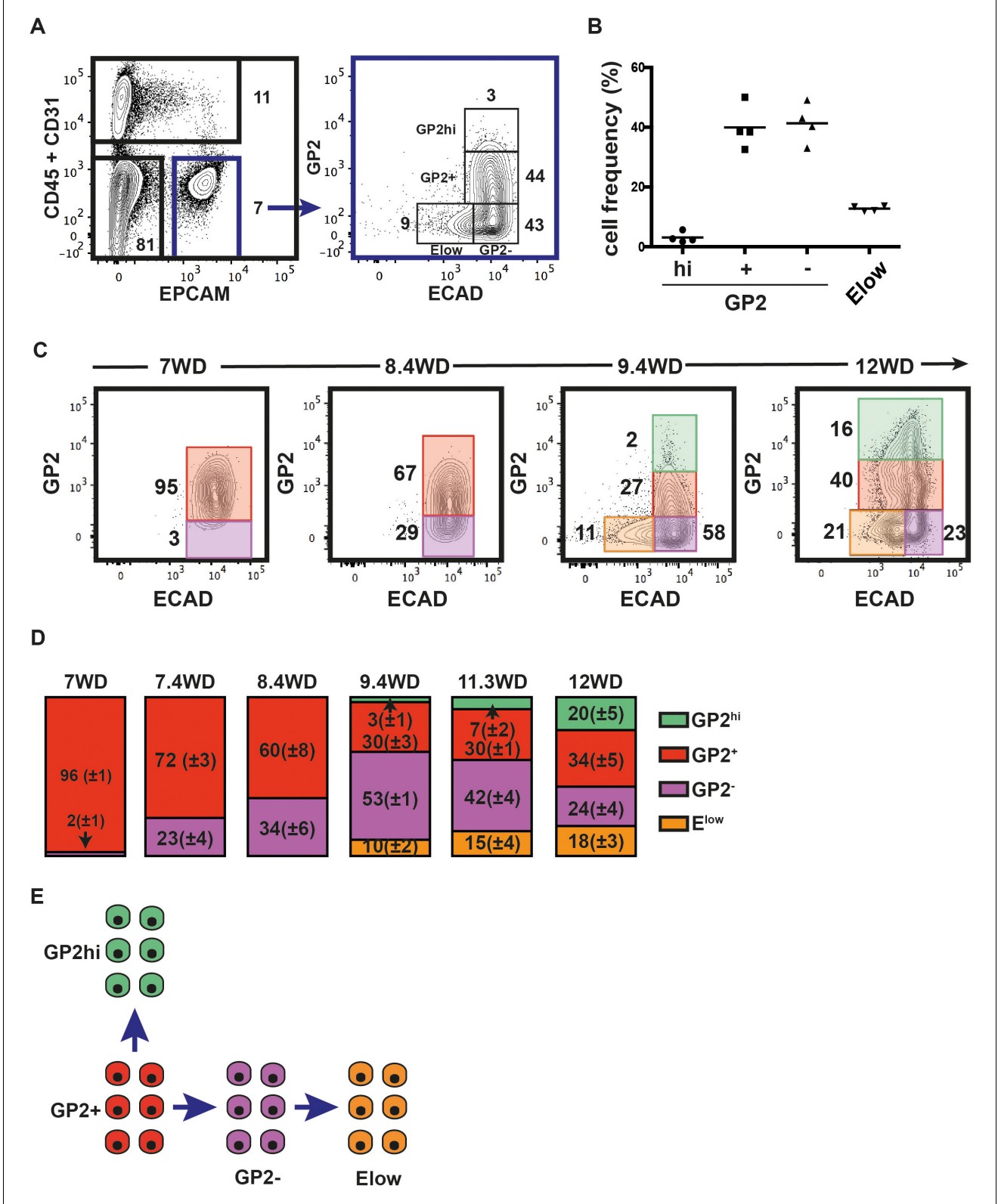

**Figure 2.** GP2 and ECAD expression in the human fetal pancreatic epithelium. GP2 and ECAD expressions were assayed by flow cytometry during development. (**A**) FACS plots display the expression at 9.4WD of CD45 and CD31 against EPCAM (left plot) and GP2 and ECAD gated on CD45⁻CD31⁻EPCAM⁺ (right plot). n = 4 (**B**) Cell frequencies of the GP2^hi (GP2^hiECAD⁺), GP2⁺ (GP2⁺ECAD⁺), GP2⁻ (GP2⁻ECAD⁺) and E^low (GP2⁻ECAD^low) populations at 9.4WD. n = 4 (mean ±SEM) (**C**) GP2 and ECAD expressions on fetal pancreases at 7-12WD gated on CD45⁻CD31⁻EPCAM⁺ cells. 7WD
*Figure 2 continued on next page*

*Figure 2 continued*

n = 2, 8.4WD n = 9, 9.4WD n = 4, 12WD n = 5. (mean ±SEM) (**D**) Cell frequencies of the GP2$^{hi}$, GP2$^{+}$, GP2$^{-}$ and E$^{low}$ populations at 7-12WD. Cell frequencies were calculated from three independent experiments for each time point. (**E**) Scheme that represents the development of GP2$^{hi}$, GP2$^{+}$, GP2$^{-}$ and E$^{low}$ populations.

The following source data and figure supplements are available for figure 2:

**Source data 1.** Cell frequency at 9.4WD by flow cytometry.
**Source data 2.** Cell frequency during development.
**Figure supplement 1.** Gating strategy for GP2 and ECAD.
**Figure supplement 2.** Expression of ECAD and GP2 during development.

## CD142 and SUSD2 reveal heterogeneity within the GP2$^{-}$ and E$^{low}$ populations during development

The GP2$^{-}$ population displayed both duct and endocrine progenitor cells markers suggesting it contains more than one cell type. We therefore sought for additional discriminant markers by scrutinizing our transcriptomic data. We observed two cell surface markers, CD142 and SUSD2 with opposite expression patterns: the E$^{low}$ population expressed lower *CD142* and higher *SUSD2* mRNA levels than the GP2$^{+}$ and GP2$^{-}$ populations (*Figure 5A*). FACS analyses showed that at 9.4WD, the GP2$^{hi}$ and GP2$^{+}$ populations were uniformly CD142$^{+}$SUSD2$^{-}$, while the GP2$^{-}$ and E$^{low}$ populations were further divided into three subsets: CD142$^{+}$SUSD2$^{-}$, CD142$^{-}$SUSD2$^{-}$ and CD142$^{-}$SUSD2$^{+}$ (*Figure 5B*). Noteworthy, the CD142$^{-}$SUSD2$^{+}$ subset was scarce in the GP2$^{-}$ population (6%), but represented 40% of the E$^{low}$ population at 9.4WD (*Figure 5B*). Accordingly, our data reflects the heterogeneity within the GP2$^{-}$ and E$^{low}$ populations that is further resolved using CD142 and SUSD2 antibodies.

Next we examined the CD142 and SUSD2 expression patterns in the GP2$^{-}$ and in the E$^{low}$ populations during development. At 7WD the majority of the GP2$^{-}$ population was CD142$^{+}$SUSD2$^{-}$ (95 ± 2%). This frequency gradually decreased to 62 ± 2% at 9.4WD while the frequency of the CD142$^{-}$SUSD2$^{-}$ subset increased from 5 ± 2% at 7WD to 43% at 11.3WD p<0.05 (*Figure 5C,D*). We detected the first SUSD2$^{+}$ cells in the GP2$^{-}$CD142$^{-}$ subset at a low frequency (1%) at 8.4WD (*Figure 5C*). The first E$^{low}$ population was detected at 8.6WD and was divided into CD142$^{+}$SUSD2$^{-}$, CD142$^{-}$SUSD2$^{-}$ and CD142$^{-}$SUSD2$^{+}$ subsets (*Figure 2—figure supplement 2A,B* and *Figure 5—figure supplement 1*). As development progressed the frequency of the E$^{low}$CD142$^{-}$SUSD2$^{+}$ subset decreased (from 48 ± 7% at 8.6WD to 10 ± 1% at 12WD; p<0.05), while the frequency of the E$^{low}$CD142$^{-}$SUSD2$^{-}$ subset increased (from 30 ± 4% at 8.6WD to 79 ± 3% at 12WD; p<0.05) (*Figure 5B,C,E*). Our data thus indicate that the first GP2$^{-}$ cells are CD142$^{+}$. They progress in their differentiation program by down-regulating CD142, then up-regulating SUSD2 and finally decreasing ECAD levels.

## Endocrine progenitors develop in the GP2$^{-}$CD142$^{-}$SUSD2$^{-}$ subset and mature within the E$^{low}$SUSD2$^{+}$ subset

We analyzed the expression pattern of endocrine markers from 8.6WD to 13WD in the three E$^{low}$ subsets. The E$^{low}$CD142$^{+}$SUSD2$^{-}$ subset did not express endocrine markers such as *CHGA* and *NEUROG3* (*Figure 6—figure supplement 1*). At 8.6WD, *CHGA, NEUROD1, NKX2-2* and *INS* were solely detected in the E$^{low}$CD142$^{-}$SUSD2$^{+}$ subset (*Figure 6A*). Interestingly, at 10-12WD, *CHGA, NEUROD1* and *NKX2-2* expressions were detected both in the E$^{low}$CD142$^{-}$SUSD2$^{+}$ and the E$^{low}$CD142$^{-}$SUSD2$^{-}$ subsets while *INS* was exclusively detected in the E$^{low}$CD142$^{-}$SUSD2$^{-}$ subset (*Figure 6B*).

Next we assessed *NEUROG3* expression in the different subsets. The first *NEUROG3$^{+}$* were detected in the GP2$^{-}$CD142$^{-}$ subset at 8.4WD before SUSD2 up-regulation (*Figure 6C*). From 8.6WD, when the E$^{low}$ population was first detected *NEUROG3* was enriched in the E$^{low}$CD142$^{-}$SUSD2$^{+}$ subset (*Figure 6C*). Our results indicate that endocrine progenitors first appear

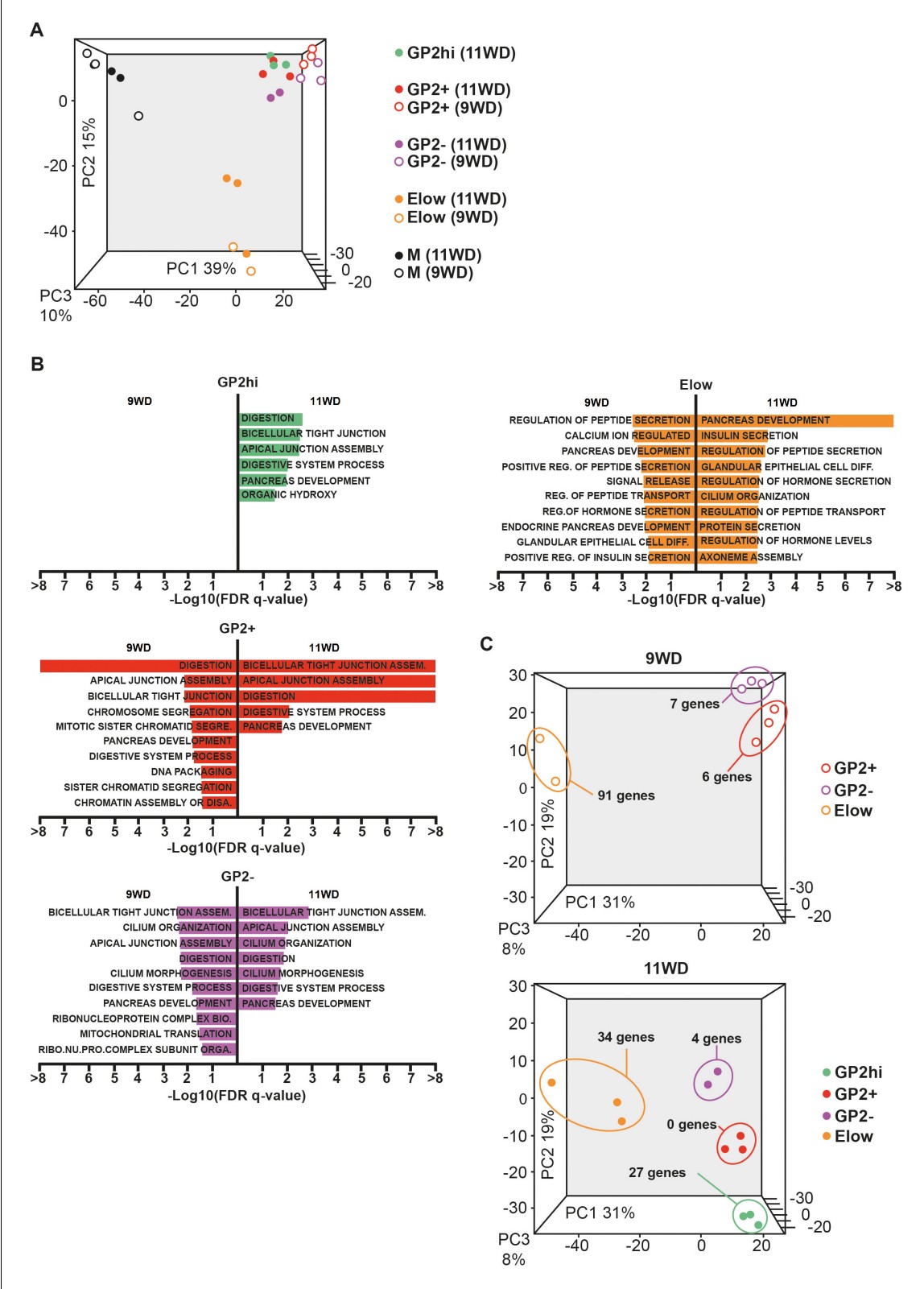

**Figure 3.** Transcriptomic analysis of the GP2hi, GP2+, GP2- and Elow populations. (**A**) PCA map of sorted pancreatic cells (epithelium and mesenchyme). (**B**) Top enriched biological processes in each cell population compared to the mesenchyme. Results were obtained with GSEA software using the GO database. (**C**) PCA map of epithelial-sorted cells (GP2hi, GP2+, GP2- and Elow populations) from 9 (top map) and 11WD (bottom map). The number of

*Figure 3 continued on next page*

*Figure 3 continued*

specifically genes enriched in each population in each population (p<0.05) is displayed. PCA maps are displayed in 2D with the three Principal Components on figure *Figure 3—figure supplement 1*) and the Gene Ontology lists in *Supplementary file 1a*.

The following figure supplements are available for figure 3:

**Figure supplement 1.** Principal component analysis.

**Figure supplement 2.** Heatmaps of genes from GO Biological processes digestion, insulin secretion and pancreas development.

**Figure supplement 3.** The GP2$^{hi}$ and E$^{low}$ populations are enriched in acinar and endocrine markers respectively.

in the GP2$^-$CD142$^-$SUSD2$^-$ subset and next mature within the E$^{low}$CD142$^-$SUSD2$^+$ subset. *INS* is first detected in the E$^{low}$ CD142$^-$SUSD2$^+$ subset and later on in the E$^{low}$CD142$^-$SUSD2$^-$ subset.

## The E$^{low}$CD142$^-$SUSD2$^+$ subset predominantly contains endocrine progenitors

To assess the frequency of cells expressing endocrine progenitor (*NEUROG3, NEUROD1, NKX2-2*) and duct (*CFTR*) markers in the GP2$^-$CD142$^-$SUSD2$^-$ and in the E$^{low}$CD142$^-$SUSD2$^+$ subsets, we performed single cell RT-qPCR analyses at 9WD. The GP2$^-$CD142$^-$SUSD2$^-$ subset was heterogeneous and composed of 78% *CFTR$^+$NEUROG3$^-$* cells, 16% *NEUROG3$^+$* and 6% *CFTR$^-$NEUROG3$^-$* cells. Among the *NEUROG3$^+$*, 47% were *NEUROD1$^-$NKX2.2$^-$*, 43% co-expressed *NEUROD1*, and 10% co-expressed *NEUROD1* and *NKX2.2*. Interestingly, 60% of *NEUROG3$^+$* were *CFTR$^+$* (*Figure 6D*). Conversely, only 2% of the E$^{low}$CD142$^-$SUSD2$^+$ subset was *CFTR$^+$NEUROG3$^-$*, while 74% was *NEUROG3$^+$* and 24% *CFTR$^-$NEUROG3$^-$*. *NEUROG3$^+$* were essentially homogenous as 100% expressed *NEUROD1$^+$* and 83% co-expressed *NKX2-2* as well (*Figure 6E*). Our findings demonstrate that the majority of the E$^{low}$CD142$^-$SUSD2$^+$ subset is *NEUROG3$^+$* co-expressing *NEUROD1* and *NKX2-2*.

## Human pluripotent stem cells differentiation into pancreatic endocrine cells mimics human fetal endocrine cell development

To determine if similar cell populations were present in pancreatic endocrine cells derived from three hPSCs (SA121 hESC, AD2.1 iPSC, AD3.1 iPSC), these cells were differentiated to the corresponding stage (stage 5) where significant endocrine induction occurs (*Figure 7A*). CD142 and ECAD characterized three distinct populations: ECAD$^+$CD142$^+$, ECAD$^+$CD142$^-$ and ECAD$^{low}$CD142$^-$ as it is the case in the human fetal pancreas (9.4WD) (*Figure 7B,C*). Similar to the human fetal pancreas, SUSD2$^+$ cells were enriched in the ECAD$^{low}$CD142$^-$ population (*Figure 7B,C*). Moreover, *NEUROG3*, *NEUROD1* and *NKX2-2* were mainly expressed in the ECAD$^{low}$CD142$^-$SUSD2$^+$ population as observed in the human fetal pancreas at 8.6WD (*Figures 6A,C* and *7D*). Finally, from stage 2–5 the temporal expression pattern of CD142 and ECAD was reminiscent of the ones occurring in the human fetal pancreas with a sharp decrease of CD142$^+$SUSD2$^-$ subsets (from 95 ± 2 of CD142$^+$-SUSD2$^-$ at stage 2 to 23 ± 4 of CD142$^+$SUSD2$^-$ subset; p<0.05) (*5D* and *7E*), as was also the case for SUSD2 expression (*Figures 5E* and *7F*) in the three hPSC lines. To conclude we demonstrate that pancreatic endocrine cells derived from human pluripotent stem cells appear to go through the same intermediate developmental stages as observed during in vivo development.

## Discussion

In this study, we reconstruct human fetal pancreatic differentiation by combining a specific combination of cell surface markers. Although a large set of data is available concerning development in rodent models, there is only limited knowledge on human development. Importantly, while rodent and human pancreatic development share many similarities, they differ on several aspects (*Scharfmann et al., 2013*; *Nair and Hebrok, 2015*). For example, the global shape and the way islet cells cluster are different between rodent and human pancreas (*Brissova et al., 2005*). Moreover, the expression pattern of major transcription factors crucial for proper pancreas development such

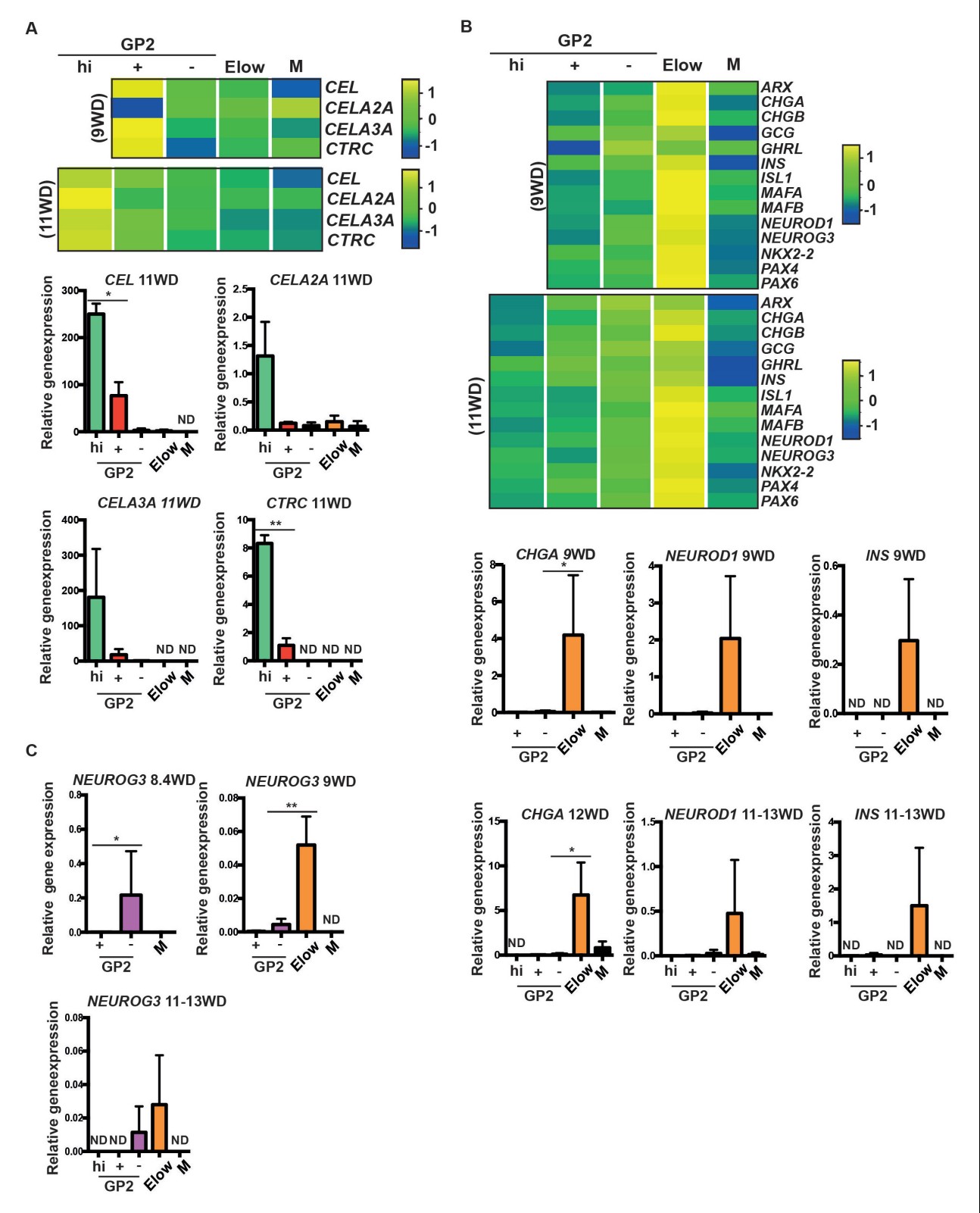

**Figure 4.** Characterization of the GP2hi, GP2+, GP2- and Elow populations. (A,B) Expression of acinar (A) and endocrine (B) markers in the GP2hi, GP2+, GP2- and Elow populations by global transcriptomic analyses and by RT-qPCR. (C) Expression of *NEUROG3* by RT-qPCR in the GP2hi, GP2+, GP2- and M populations. Heat maps and RT-qPCR are representative of 3 independents experiments. ND = Non Detected. M = CD45-CD31-EPCAM-. *p<0.05, **p<0.001, *t* test. (mean ± SEM).

*Figure 4 continued on next page*

*Figure 4 continued*

The following figure supplement is available for figure 4:

**Figure supplement 1.** Expression of ductal markers in the GP2$^{hi}$, GP2$^{+}$, GP2$^{-}$ and E$^{low}$ populations.

as NEUROG3, NKX2-2 and PDX1 differ between rodent and human (*Villasenor et al., 2008*; *Salisbury et al., 2014*; *Jennings et al., 2013*; *Sussel et al., 1998*; *Ohlsson et al., 1993*; *Heimberg et al., 2000*). As an example, NEUROG3 is expressed in two waves during rodent fetal pancreatic development (*Villasenor et al., 2008*) while only a single-phase is observed in human (*Jennings et al., 2013*). It can be speculated that the number of differences between rodents and humans are under-estimated and more will be discovered by studying human pancreatic development. This knowledge will be essential to better design protocols to direct pluripotent stem cells towards functional insulin-producing pancreatic beta cells (*Pagliuca et al., 2014*; *Rezania et al., 2014*; *Russ et al., 2015*).

Major progress has recently been made in the field of hPSC differentiation and a number of different cell types can be efficiently generated either in vivo following transplantation into immune-incompetent mice or in vitro under controlled conditions. However, the cells generated in vitro from hPSC do not seem fully differentiated and in some cases, display a fetal phenotype rather than an adult one. This is the case for hepatocytes (*Baxter et al., 2015*), cardiomyocytes (*Karakikes et al., 2015*), neurons (*Playne and Connor, 2017*) and pancreatic beta cells (*Hrvatin et al., 2014*). These partial successes could be in part due to the limited knowledge on human cell development. Here, we specifically designed new approaches to dissect in great detail pancreatic differentiation in human. We based our study on human fetal pancreases from 7 to 13WD. This developmental period corresponds to E12.5-E17 in the mouse when pancreatic progenitors first proliferate and next develop into differentiated cells (*Gittes, 2009*; *Jennings et al., 2015*). At 6-7WD, the human pancreatic epithelium is mainly composed of PDX1$^{+}$NKX6-1$^{+}$ pancreatic progenitors (*Cebola et al., 2015*; *Riedel et al., 2012*) while endocrine cells are extremely rare (*Polak et al., 2000*; *Castaing et al., 2001*). During the following weeks, proliferating epithelial pancreatic progenitors differentiate into endocrine, acinar and duct cells (*Jennings et al., 2013*; *Capito et al., 2013*).

The fetal pancreas is a compound organ composed of epithelial, mesenchymal, endothelial and hematopoietic cells. We excluded endothelial and hematopoietic cells using CD31 and CD45 antibodies and used the transmembrane glycoprotein EPCAM as a marker to segregate the fetal pancreatic epithelium from the mesenchyme. Previous data indicated that EPCAM is expressed during human fetal life (18-20WD) in the ductal pancreatic epithelium and in developing islet-like cells but also in the adult human pancreas in duct and islet cells (*Cirulli et al., 1998*). Based on this previous report we demonstrated that EPCAM also marks the fetal pancreatic epithelium at earlier developmental stages, between 7 and 12WD. Moreover, we demonstrated that the EPCAM$^{+}$ compartment contains PDX1$^{+}$NKX6-1$^{+}$ double-positive cells, a hallmark of pancreatic progenitors (*Jennings et al., 2013*; *Cebola et al., 2015*). Our data also indicate that 45% of the EPCAM$^{+}$ compartment expresses lower levels of PDX1 and NKX6-1. It would be interesting to determine if these cells are upstream progenitors of EPCAM$^{+}$PDX1$^{+}$NKX6-1$^{+}$ cells as recently suggested (*Ameri et al., 2017*). Then we further segregated the fetal pancreatic epithelium into four distinct populations using the cell surface markers GP2 and ECAD. GP2 is a glycoprotein that is highly enriched in the acinar cells of the adult pancreas (*Hoops and Rindler, 1991*; *Yu et al., 2004*). Limited knowledge is available on GP2 expression during pancreatic development. Very recently, using a model of pluripotent stem cells differentiation into pancreatic endocrine cells, GP2 was identified as a novel cell surface marker of human pancreatic progenitors (*Ameri et al., 2017*). Our ex vivo data further support this claim. First, we observed that at 7WD, nearly all pancreatic epithelial cells are GP2$^{+}$. Moreover, as development proceeds, the frequency of GP2$^{+}$ cells decreases. Finally, transcriptomic profiling performed at different developmental stages strongly suggests that the GP2$^{+}$ population can differentiate into endocrine and exocrine cells. Collectively, these data indicate that in the early human fetal pancreas, GP2 is indeed a cell surface marker of a multipotent cell population. Although the GP2$^{+}$ cell population is multipotent, it is not yet certain if the GP2$^{+}$ population contains multipotent GP2$^{+}$ progenitor that

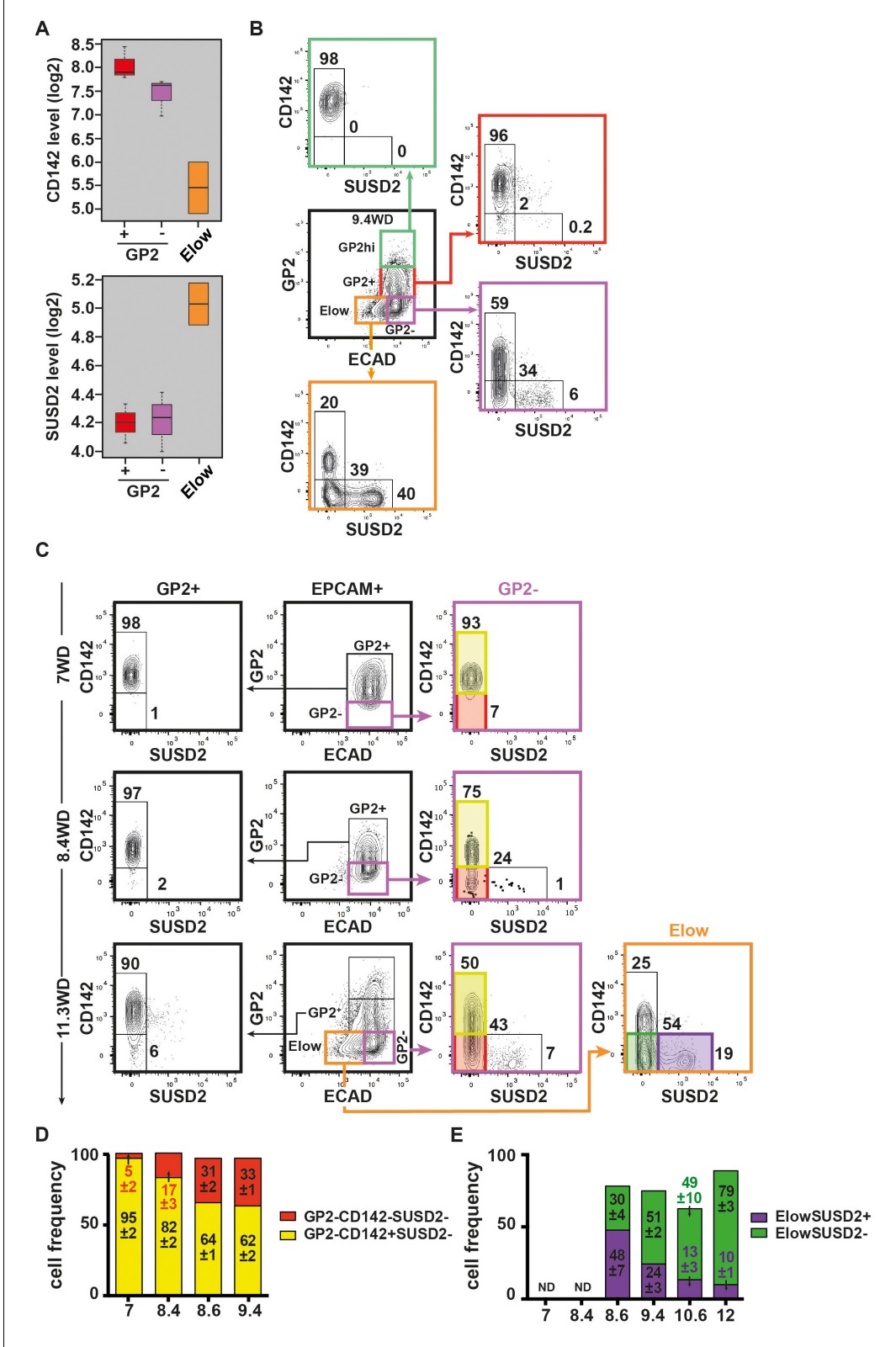

**Figure 5.** Expression of CD142 and SUSD2 in the GP2hi, GP2+, GP2- and Elow populations. (**A**) Expression of *CD142* and *SUSD2* in GP2+, GP2- and Elow populations at 9WD by microarray analysis. Boxplots were obtained using standardized log2 expression values. (**B**) Expression of CD142 and SUSD2 in the GP2hi (in green), GP2+ (in red), GP2- (in purple) and Elow (in orange) populations at 9.4WD by flow cytometry. GP2 and ECAD expressions were gated on live CD45-CD31-EPCAM+ cells. (**C**) Expression of CD142 and SUSD2 in the GP2+ (left plot), GP2- (right plot) and Elow (far right plot)

*Figure 5 continued on next page*

*Figure 5 continued*

populations at 7WD, 8.4WD, and 11.3WD. (D) Cell frequencies of the GP2⁻CD142⁺SUSD2⁻ (named CD142⁺SUSD2⁻ in yellow) and the GP2⁻CD142⁻SUSD2⁻ (CD142⁻SUSD2⁻ in red) subsets from 7WD to 9.4WD. (mean ± SEM). (E) Cell frequencies of the E$^{low}$GP2⁻CD142⁻SUSD2⁺ (named E$^{low}$SUSD2⁺, in purple) and the E$^{low}$GP2⁻CD142⁻SUSD2⁻ (E$^{low}$SUSD2⁻, in green) subsets from 7 to 12WD. (mean ± SEM). (A) n = 2–3, (B) n = 3; C) 7WD n = 2, 8.4WD to 9.4WD n = 3. (D) n = 3.

The following figure supplement is available for figure 5:

**Figure supplement 1.** Expression of CD142 and SUSD2 in the GP2⁺, GP2⁻ and E$^{low}$ populations at 8.6WD.

can differentiate into the three lineages (endocrine, acinar and ductal) or represent a mixture of GP2⁺ progenitors already committed to a specific lineage. This could be addressed by utilizing a single cell culture approach. However, our attempts so far to culture so few cells sorted from human fetal pancreas have failed. Discovering new and efficient culture conditions such as co-culture on feeder layers could alleviate this issue (*Trott et al., 2017*).

In animal models, cell adhesion molecules have emerged as key regulators of embryonic morphogenesis and this topic has been extensively studied using pancreatic development as a model organ (*Semb, 2004*). As an example, during mouse and chick development, Pdx1⁺Nkx6.1⁺ progenitors express high levels of ECAD. NEUROG3⁺ endocrine progenitors will develop from such pancreatic progenitors while lowering their ECAD level during their delamination from the ductal tree to develop into pancreatic endocrine cells (*Gouzi et al., 2011*). Information remains scarcer on the regulation of ECAD levels during human development. Our results indicate that at 7WD, GP2⁺ pancreatic progenitors express ECAD at high levels. At 8.6WD, ECAD$^{low}$ cells appear, their frequency increasing while development progresses. Interestingly, NEUROG3-expressing cells first appear at 8.4WD in the ECAD⁺ population and are found later on in the ECAD$^{low}$ population. Whether it is linked to their delamination remains to be demonstrated. This step is followed (at 8.6WD) by the expression of endocrine markers such as *CHGA*, *NEUROD1* and *INS*. Of importance, endocrine cells expressed low but significant levels of ECAD. This perfectly fits with mouse data that indicate that during development ECAD function is necessary for proper aggregation of endocrine cells after delamination (*Dahl et al., 1996*). Taken together, our data demonstrate that ECAD levels are tightly regulated during specific steps of human pancreatic development.

We further refined specific pancreatic cell populations by using CD142 and SUSD2 found in our transcriptomic analysis as additional cell surface markers. CD142 has been proposed as a marker of pancreatic endodermal cells that also labels additional cell types (*Kelly et al., 2011*). SUSD2 was previously used as a marker to enrich *NEUROG3⁺* from hPSC derived pancreatic cells and the human fetal pancreas (*Liu et al., 2014*). Here, with our set of markers we reconstructed human pancreatic cell differentiation (*Figure 7G*). The full combination of markers was required for this reconstruction as none of the markers was specific to the different subsets (*Figure 7—figure supplement 1*). Our data indicate that GP2⁺CD142⁺ pancreatic progenitors can either give rise to GP2$^{hi}$CD142⁺ acinar cells or enter the endocrine pathway and express *NEUROG3* by turning off GP2 and CD142. Endocrine maturation further progresses by up-regulating SUSD2 and decreasing ECAD level (*Figure 7G*). The first *INS*⁺ cells were detected at 8.6WD in the E$^{low}$ population as SUSD2⁺ and later on as SUSD2⁻. Recently, differences in gene expression and functionality were observed between fetal, neonatal and adult beta cells (*Hrvatin et al., 2014*; *Jermendy et al., 2011*; *Blum et al., 2012*). Moreover, heterogeneity between adult beta cells was also recently described (*Dorrell et al., 2016*; *Bader et al., 2016*). Whether *INS*⁺ cells positive for SUSD2 represent a first wave of beta cells that could be poly-hormonal remains to be tested. Moreover, comparative analyses between *INS*⁺ cells from E$^{low}$SUSD2⁺ and E$^{low}$SUSD2⁻ subsets will define the differences in their transcriptional factor networks.

FACS-based approaches have been used since the eighties in the hematopoietic field to dissect hematopoiesis (*Spangrude et al., 1988*). However, it has rarely been used in the field of pancreatic development and even less frequently with human fetal pancreases. In the pancreas, the majority of the cell sorting approaches was performed by analyzing fluorescent signals from tagged proteins derived from transgenic mice (*Gu et al., 2004*; *Miyatsuka et al., 2014*). While highly informative, this strategy cannot be used to dissect human pancreatic development. Antibodies against cell

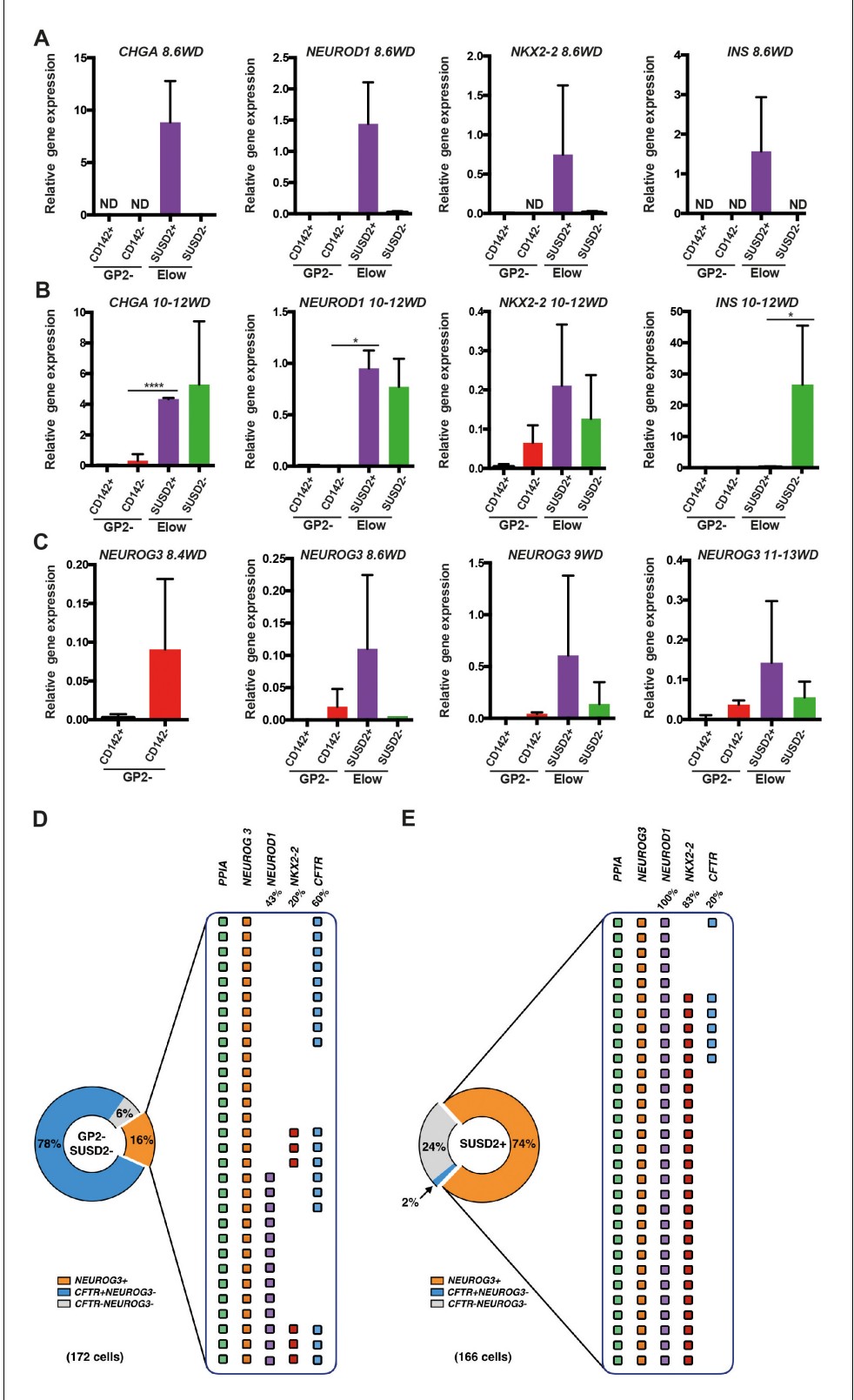

**Figure 6.** Molecular characterization of the GP2⁻CD142⁻SUSD2⁻ and E^lowGP2⁻CD142⁻SUSD2⁻ subsets. (**A, B**) RT-qPCR for *CHGA, NEUROD1, NKX2-2* and *INS* on sorted GP2⁻CD142⁺SUSD2⁻ (GP2⁻CD142⁺), GP2⁻CD142⁻SUSD2⁻ (GP2⁻CD142⁻), E^lowGP2⁻CD142⁻SUSD2⁺ (E^lowSUSD2⁺) and E^lowGP2⁻CD142⁻SUSD2⁻ (E^lowSUSD2⁻) subsets at 8.6 and 10-12WD. (**C**) RT-qPCR for *NEUROG3* at 8.4, 8.6, 9 and 11-13WD in the GP2⁻CD142⁺SUSD2⁻ (GP2⁻CD142⁺), GP2⁻CD142⁻SUSD2⁻ (GP2⁻CD142⁻), E^lowGP2⁻CD142⁻SUSD2⁺ (E^lowSUSD2⁺) and E^lowGP2⁻CD142⁻SUSD2⁻ (E^lowSUSD2⁻) subsets. (**D, E**) Single cell RT-qPCR

*Figure 6 continued on next page*

*Figure 6 continued*

at 9WD on 172 GP2$^-$CD142$^+$SUSD2$^-$ (GP2$^-$SUSD2$^-$, left panel) and 166 E$^{low}$GP2$^-$CD142$^-$SUSD2$^+$ (SUSD2$^+$, right panel) cells for the expression of *PPIA*, *NEUROG3*, *NEUROD1*, *NKX2-2* and *CFTR*. Pie charts represent the percentage of *NEUROG3$^+$* (in orange), *NEUROG3$^-$CFTR$^+$* (in blue) and *NEUROG3$^-$CFTR$^-$* cells (in grey). For *NEUROG3$^+$* the percentages of *NEUROD1$^+$*, *NKX2-2$^+$*, and *CFTR$^+$* cells are displayed. Each line represents one cell. *PPIA$^-$* cells were excluded from the analysis. (A–C) n = 3, (D, E) n = 2. *p<0.05, ****p<0.0001, *t* test. (mean ±SEM).

The following figure supplement is available for figure 6:

**Figure supplement 1.** Expression of *CHGA* and *NEUROG3* in the E$^{low}$CD142$^+$ subsets.

surface markers were used in a limited number of studies on pancreas development and mainly in rodents. It was found that a combination of CD49f and CD133 antibodies can be used to enrich fraction in mouse pancreatic progenitors expressing *NEUROG3*-expressing cells with some data on human fetal pancreas. However, the enrichment in *NEUROG3* on human fetal pancreas was limited using this combination of markers (*Sugiyama et al., 2007*). More recently, SUSD2 was used as a marker to enrich NEUROG3$^+$ cells in hPSC -derived pancreatic cells and in the human fetal pancreas (*Liu et al., 2014*). Our data confirm this point. However our single cell qPCR indicate that SUSD2 does not mark all, but only a subset of *NEUROG3$^+$* that co-expressed *NEUROD1* and *NKX2-2*. Thus, by using a combination of antibodies against cell surface markers, we demonstrate that cell populations highly enriched in specific functions can be sorted from the human fetal pancreas at different stages of development, allowing the reconstruction of the differentiation program. Our model constitutes a key advancement in understanding human fetal pancreas development by mapping out the pattern of differentiation of the three main pancreatic lineages. We further refined our work for the endocrine pathway by describing for the first time, discrete stages of human pancreatic endocrine cell differentiation and showed that our development model also applies to the in vitro differentiation of hPSCs into pancreatic endocrine cells. To the best of our knowledge, this type of side-by-side comparison that demonstrates that in vitro hPSC differentiation mimics in vivo events has rarely been done.

In conclusion, we provide a novel way of approaching human pancreatic differentiation. Our work will be useful to fill the limited knowledge on human pancreas development. It should also pave the way for developing new cell therapies for diabetic patients.

## Materials and methods

### Pancreatic dissection and cell suspension preparation

All experiments on human fetal pancreas were performed at INSERM Paris, France. Human fetal pancreases were isolated from surgical abortion done by suction aspiration between 7 to 13 weeks of development (*Castaing et al., 2001*; *Capito et al., 2013*; *Scharfmann et al., 2014*) in compliance with the French bioethics legislation and the guidelines of our institution. Approval was obtained from Agence de Biomedecine, the French competent authority along with maternal written consent. Ages were determined on the basis of time since the last menstrual period and hand and foot morphology. Fetal pancreases were micro dissected with forceps under a binocular magnifying lens, rinsed with Hanks Balanced Salt Solution (HBSS) from Gibco to remove contaminating blood cells and gently disrupted using forceps. Afterwards, pancreases were incubated for 5 min in collagenase V (0.5 mg/ml) (Sigma Aldrich) in HBSS in the presence of calcium and magnesium. Cells were rinsed in HBSS and then incubated for 5 min in trypsin (0.05%) (Gibco). Finally, cells were rinsed in HBSS supplemented with 20% Fetal calf serum (from Eurobio). Pluripotent stem cells were washed with PBS without Ca2$^+$ and Mg2$^+$ (Invitrogen, FRANCE) and incubated with TrypLE select for 1–3 min.

### Maintenance and differentiation of human pluripotent stem cell lines

A human ESC (SA121) (*Heins et al., 2004*) obtained from Takara and two iPSC (SB Ad2.1 and SB3.1) (*van de Bunt et al., 2016*) obtained from the StemBANCC consortium were applied for this study. These cell lines had been confirmed to be pluripotent by evaluation of pluripotency marker expression, tri-lineage differentiation and karyotyping and tested negative for mycoplasma contamination.

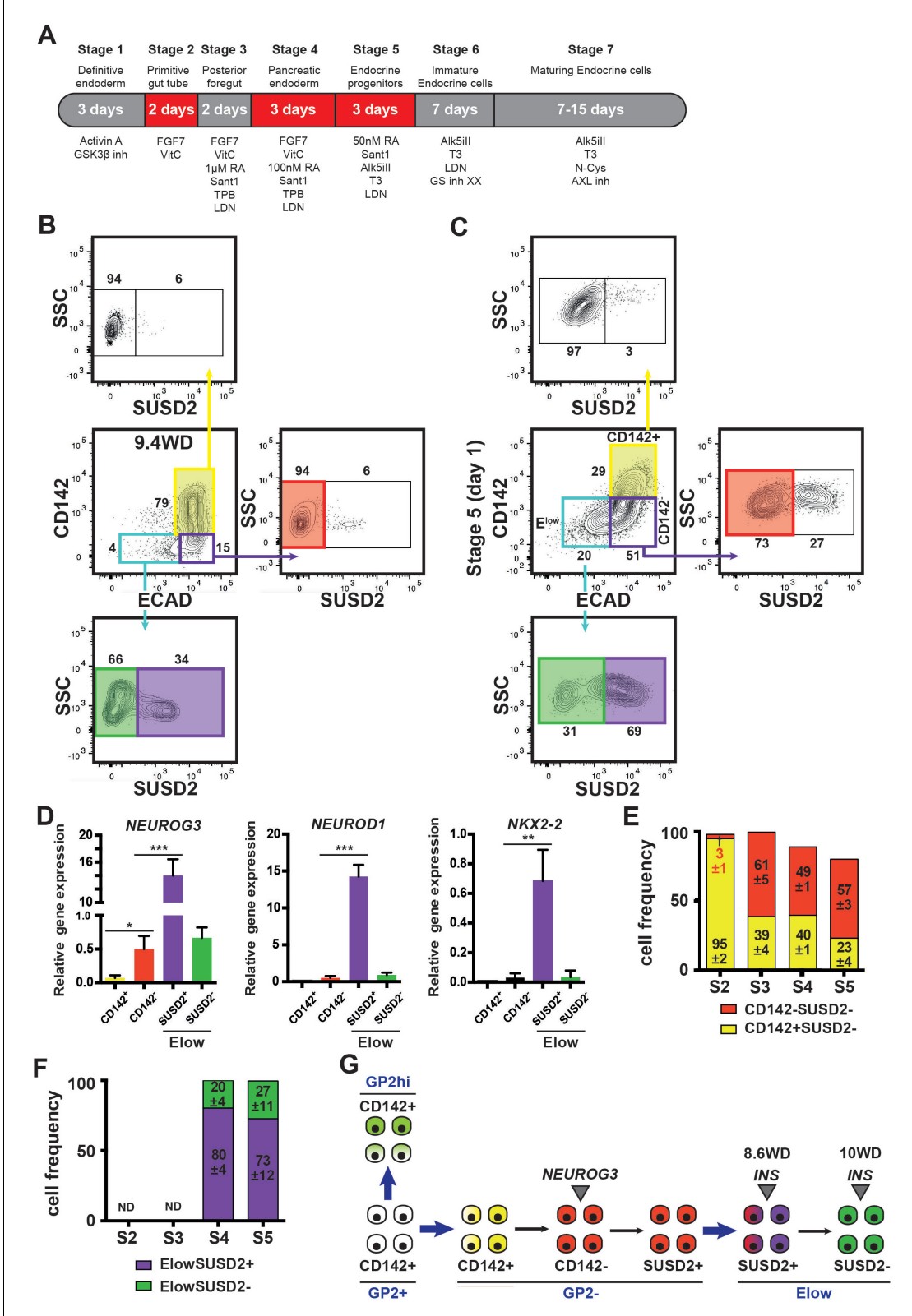

**Figure 7.** Expression of CD142, ECAD and SUSD2 in pancreatic endocrine cells derived from hPSCs. (**A**) Scheme of in vitro differentiation of hPSCs into endocrine pancreatic cells. (**B, C**) Expression of CD142, ECAD (gated on lived CD45[-]CD31[-]EPCAM[+] cells for the human) and SUSD2 in CD142[+], CD142[-] and E[low] population by flow cytometry in the human fetal pancreas at 9.4WD (in B) and in pancreatic endocrine cells derived from hPSCs (AD3.1 iPSC) at stage 5 (in C). (**D**) RT-qPCR on CD142[+]E-CAD[+]SUSD2[-] (named CD142[+]), CD142[-]E-CAD[+]SUSD2[-] (named CD142[-]), CD142[-]E-CAD[low]SUSD2[+] (named

*Figure 7 continued on next page*

*Figure 7 continued*

E$^{low}$SUSD2$^{+}$), CD142$^{-}$E-CAD$^{low}$SUSD2$^{-}$ (named E$^{low}$SUSD2$^{-}$) from hPSCs (AD3.1 iPSC) at stage five for *NEUROG3, NEUROD1* and *NKX2-2*). (**E**) Cell frequencies of CD142$^{+}$SUSD2$^{-}$ (in yellow) and CD142$^{-}$SUSD2$^{-}$ (in red) from hPSCs (SA121 hESC, AD2.1 iPSC, AD3.1 iPSC) from stage 2–5. (**F**) Cell frequencies of E$^{low}$SUSD2$^{+}$ and E$^{low}$SUSD2$^{-}$ from hPSCs (SA121 hESC, AD2.1 iPSC, AD3.1 iPSC) from stage 2–5. (**G**) Scheme representing human pancreatic differentiation across development using cell surface markers. (**B–F**) n = 3. *p<0.05, **p<0.001, ***p<0.005, *t* test.

The following figure supplement is available for figure 7:

**Figure supplement 1.** Expression of CD142 and SUSD2 in the mesenchyme, the endothelial/hematopoietic and the epithelial compartments.

All three lines were cultured in mTeSR1 medium (StemCell Technologies) on hESC-qualified matrigel (Corning). Cells were passaged every 3–4 days or when confluent by dissociating to a single cell solution using TrypLE select (ThermoFisher). Single cells were seeded onto freshly coated Matrigel tissue culture flask in mTeSR1 containing of 5 uM Tiger (Rock inhibitor, Sigma-Aldrich) and medium was replenished daily. For differentiation of the hESC/hiPSC lines cells were dissociated to a single cell solution using TrypLE select and resuspended mTeSR1 with 5 uM Tiger. Cells were seeded at a concentration of $0.35 \times 10^6$ cells/cm$^2$ onto growth-factor reduced matrigel CellBIND surfaces (Corning). Cells were incubated for 24 hr before the start of the differentiation. Differentiation to the pancreatic lineage was conducted as described in a previously published protocol (*Rezania et al., 2014*) with the following modifications: CHIR99201 (Axon Medchem) was applied at 3 uM and 0.3 uM concentration for the first and second day, respectively of the definitive endoderm differentiation instead of MCX-928 used in the original protocol. MCDB131 medium (Life technologies) was used as basal medium throughout the differentiation instead of BLAR medium. Cells were not dissociated and incubated as clusters on air-liquid filters during stage 5; instead cells were kept in 2D cultures throughout the differentiation.

## Flow cytometry

Following dissociation, cells were incubated with antibodies for 20 min in FACS medium (HBSS +2% FCS), then rinsed in FACS medium and re-suspended in FACS medium with Propidium Iodide (1/ 4000) (Sigma Aldrich) or DAPI solution (0,1 ug/ml) (BD Biosciences) to label dead cells. For intra-cellular staining cells were fixed for 5 min in 3% PFA and then rinsed in DPBS from GIBCO. Then cells were permeabilized in DPBS +3% BSA +0.3% Triton and incubated with antibodies overnight at 4°C, rinsed and re-suspended in DPBS. For each antibody, optimal dilution was determined by titration. The following antibodies were used: anti CD45-PerCP/Cy5.5 (1/20, clone 2D1, Biolegend; RRID: AB_2566351), anti CD31-PerCP/Cy5.5 (1/20, clone WM59, Biolegend; RRID: AB_2566174), anti-EPCAM-Brillant violet 605 (1/20, clone 9C4, Biolegend; RRID: AB_2562518), anti ECAD-PE-Cy7 (1/20, clone 67A4, Biolegend; RRID: AB_2563096), anti-GP2-PE (1/5, clone 3G7H9, MBL; RRID: AB_11160953), anti-SUSD2-VioBrightFITC (1/20, W5C5, Miltenyi Biotec; RRID: AB_2653618), anti-CD142-VioBlue (1/ 20, clone HTF-1, Miltenyi Biotec; RRID: AB_2655132), anti-PDX1-alexaFluor 488 (1/20, clone 658A5, BD Biosciences) and anti-NKX6-1-alexaFluor 647 (1/20, clone R11-560, BD Biosciences). An ARIA III (BD Bioscience) was used for cell sorting and a FACS LSRFortessa for analysis (BD Bioscience). Data were analyzed using FlowJo 10.2 software. Dead cells were excluded from analyses performed on lived cells. Lived cells referred to cells that did no incorporate the dead cell stain Propidium Iodide.

## Immunohistochemistry

Human fetal pancreatic sections (4–5 μm thick) were prepared and processed as previously described (*Castaing et al., 2005*). The following primary antibodies were used anti-mouse EPCAM (1:500, clone VU1D9, Cell Signaling) and anti-rabbit PDX1 (1/1000) (*Duvillié et al., 2003*). The secondary antibodies were anti-mouse Alexa Fluor 488 antibodies (1:400, Life Technologies) and anti-rabbit Alexa Fluor 594 antibodies (1:400, Jackson ImmunoResearch). The nuclei were stained using the Hoechst 33342 fluorescent stain (0.3 mg/ml, Invitrogen, France).

## Bulk and single cells RT-qPCR

For human fetal pancreas, single to 100 cells were sorted in 9 µL of RT/pre-amp mix from the One-Step qRT-PCR Kit (Invitrogen, France). Pre-amplified (20 cycles) cDNA was obtained according to manufacturer's notice and was used for qPCR reaction. The pre-amplified cDNA was then used for qPCR using the TaqMan protocol. RT, pre-amplification and qPCR were performed using TaqMan primers from Applied Biosystems. The following TaqMan primers were used: *PPIA* (Hs04194521_s1), *HPRT1* (Hs99999909_m1), *PDX1* (Hs00236830_m1 and Hs00426216_m1), *NKX6-1* (Hs00232355_m1), *CEL* (Hs01068709_m1), *CELA2a* (Hs04194660_s1), *CELA3A* (Hs00371667_gH), *CTRC* (Hs00200713_m1), *SOX9* (Hs00165814_m1), *NEUROG3* (Hs01875204_s1), *CHGA* (Hs00900370_m1), *NEUROD1* (Hs01922995_s1), *INS* (Hs02741908), *NKX2-2* (Hs00159616_m1). RT-qPCR results are presented in arbitrary units (AU) relative to expression of the control gene *PPIA* (for human qPCR) and *HPRT* (for hPSC qPCRs). All qPCRs on the human fetal pancreas were assayed using *PPIA* while *HPRT* was used for the qPCR on hPSCs. QPCRs were run on QuantStudio from ThermoFischer and single cell qPCR were assayed on the Biomark from Fluidigm following manufacture's instructions.

For human pluripotent stem cell differentiation cells were sorted from three independent differentiations of the ADSB3.1 iPSC line. Cells were lysed in RP1 lysis buffer from the NucleoSpin RNA/Protein purification kit (Macherey-Nagel) and stored at −80°C until purification of RNA. RNAs were extracted using NucleoSpin RNA/protein purification kit. RNA was subsequently converted to cDNA using iScript cDNA synthesis kit (Biorad) according to manufactures instructions. RT-qPCR was performed on a MX3005P qPCR system (Agilent Genomics) using a fast-2-step protocol (first 95°C for 1 min, then 40 cycles of 95°C for 10 s, 60°C for 25 s). The following TagMan primers were used: *NEUROG3* (Hs01875204_s1), *NEUROD1* (Hs01922995_s1), *NKX2-2* (Hs00159616_m1), *bActin* (Hs01060665_g1) and *HPRT* (Hs99999909_m1).

## Global transcriptomic analysis

Total RNA was obtained from Trizol-preserved (Invitrogen, France) sorted cells by chloroform extraction, quantified and assessed for quality by Agilent-2100 Bioanalyzer (Agilent, Santa Clara, CA). cDNAs were prepared and amplified using Ovation Pico WTA System V2 kit (NuGEN Technologies, San Carlos, CA, USA) and hybridized onto GeneChip Human Gene 2.0 ST Array (Affymetrix,Santa Clara, CA, USA). Quality Control and normalization (RMA) were conducted using Bioconductor Project software (https://www.bioconductor.org), which provides log2 transformed expression values. Data were extracted from raw CEL files using a custom GeneChip library file (CDF file) provided by http://brainarray.mbni.med.umich.edu/CustomCDF (*Dai et al., 2005*).

## Transcriptomic statistical analysis

Two-sample comparisons were made using two-tailed Student t-test. Transcripts for which any p-value was above 0.05 were filtered out, leaving 6444 transcripts for further analyses. We compared expression of these genes in each cell population using Gene Set Enrichment Analysis (GSEA) software and the 'Biological Process' database of the Gene Ontology Consortium (*Subramanian et al., 2005*). Biological processes with FDR q-value <0.05 were considered significant.

### Cut off to define the 'specific enriched genes':

Each population (GP2$^{hi}$, GP2$^+$, GP2$^-$, E$^{low}$ and M) was compared to the other populations using two-tailed Student t-test. Genes were considered as enriched in a specific population when over-expressed (p-value<0,05 and Fold Change > 2) in a population compared to any of the others. The list of the 'specific enriched genes' is displayed in *supplementary file 1b and c*.

### Expression pattern of the enriched fetal genes in the adult pancreas:

We use data from the single-cell RNA-seq (*Segerstolpe et al., 2016*). The normalized data are available on ArrayExpress (http://www.ebi.ac.uk/arrayexpress/ experiments/E-MTAB-5060). Next, we generated heatmaps displaying the expression of the fetal 'specific enriched genes' in the adult acinar, alpha, beta, epsilon, gamma and ductal cells.

Heatmaps were generated by the 'heatmap2' function from gplots R package (https://cran.r-project.org/web/packages/gplots/) on standardized log2 expression values, with Pearson correlation as the distance function.

## Acknowledgements

We would like to thank Delphine Bredel (INSERM U1016) for technical assistance and Ana Cumano (INSERM U668) for supports. This work was supported by the HumEn project funded by the European Commotion's Seventh Framework Programme for Research, (agreement No 602587) (RS and HS) and the Foundation Bettencourt Schueller (to RS). The RS laboratory belongs to the Laboratoire d'Excellence consortium Revive and to the Departement Hospitalo-Universitaire (DHU) Autoimmune and Hormonal disease. The research leading to the results on the hiPSC has received support from the Innovative Medicines Initiative Joint Undertaking under grant agreement n° 115439, resources of which are composed of financial contribution from the European Union's Seventh Framework Programme (FP7/2007-2013) and EFPIA companies' in kind contribution. This publication reflects only the authors' views and neither the IMI JU nor EFPIA nor the European Commission are liable for any use that may be made of the information contained therein. Work on the hESC line was done without the support from StemBANCC.

## Additional information

### Funding

| Funder | Grant reference number | Author |
| --- | --- | --- |
| Seventh Framework Programme | 602587 | Raphaël Scharfmann |
| Innovative Medicines Initiative | 115439 | Raphaël Scharfmann |

The funders had no role in study design, data collection and interpretation, or the decision to submit the work for publication.

### Author contributions

CR, Conceptualization, Formal analysis, Investigation, Methodology, Writing—original draft; NG, Formal analysis, Methodology; CB, Investigation, Methodology; JA, Conceptualization, Writing—review and editing; JSK, Investigation, Writing—review and editing; MH, Supervision, Funding acquisition, Writing—review and editing; CH, Investigation, Methodology, Writing—review and editing; HS, Conceptualization, Supervision, Writing—review and editing; RS, Conceptualization, Funding acquisition, Writing—original draft

### Author ORCIDs

Raphaël Scharfmann, http://orcid.org/0000-0001-7619-337X

## Additional files

### Supplementary files

• Supplementary file 1. Gene Ontology list. (a) List of gene sets and corresponding genes used in GSEA analysis displayed in *Figure 3B*. Gene sets were obtained from the gene ontology database. (b) List of differentially expressed genes specific to each population at 9WD. E$^{low}$: in yellow, GP2$^-$: in purple, GP2$^+$: in red, M: white. Values are displayed in Log2. (c) List of differentially expressed genes specific to each population at 11WD. E$^{low}$: in yellow, GP2$^-$: in purple, GP2$^+$: in red, M: white. Values are displayed in Log2.

### Major datasets

The following dataset was generated:

| Author(s) | Year | Dataset title | Dataset URL | Database, license, and accessibility information |
| --- | --- | --- | --- | --- |
| Ramond C, Glaser N, Berthault C, | 2017 | Expression data from cells sorted from human fetal pancreas | https://www.ncbi.nlm.nih.gov/geo/query/acc. | Publicly available at NCBIGene Expression |

Ameri J, Kirkegaard JS, Hansson M, Honoré C, Semb H, Scharfmann R

cgi?acc=GSE96697

Omnibus (accession no: GSE96697)

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

## Appendix 1

# Methods supplementary information

## Determination of the developmental stage

Weeks of development (WD) were determined based on the length and the anatomy of the foot and the hand. We also used the Human Embryo Resource website (https://embryology.med.unsw.edu.au/embryology/index.php/Embryonic_Development). Then, the precise age was calculated with the length of the foot and the following formula: Age = 4.11 X Foot Length + 5.60 (*Manjunatha et al., 2012*) from the Egyptian Journal of Forensic Sciences).

## Replicates

All replicates in this study are biological replicates. The 'n' in the figure legends indicates the numbers of independent biological replicates at each developmental stage. Each flow cytometry plot displays the results from a single human fetal pancreas. An unpaired Student's t-test was performed on the FACS data of *Figures 2D*, *5D–E* and *7E*. Below is the number of independent pancreas analyzed:

- Figure 1A: 9 pancreases at 9.7WD
- Figure 1C-D: 3 pancreases at 9WD
- Figure 2A: 4 pancreases at 9.4WD
- Figure 2C: 2 pancreases at 7WD, 9 pancreases at 8.4WD, 4 pancreases at 9.4WD, 5 pancreases at 12WD
- Figure 5B: 3 pancreases at 9.4WD
- Figure 5C: 2 pancreases at 7WD, 3 pancreases at 8.4WD, 3 pancreases at 11.3WD.
- Figure 7B: 3 pancreases at 9.4WD
- Figure 7C: three independent FACS analyses assayed on AD3.1 cell line differentiated until stage 5 day 1.
- Figure 7E-F: three independent FACS analysis assayed on three different hPSC lines differentiated from stage 2 to 5: SA121 hESC, AD2.1 iPSC, AD3.1 iPSC. Figures display the data from the three lines.

The variability between individual fetal pancreases at the same WD is displayed in *Figures 2D*, *5D–E* and *7E–F*.

## qPCR

All qPCR are from three independent experiments derived from the sorting of three independent fetal pancreases.

## Global transcriptomic analysis

We performed a global transcriptomic analyses on three independent pancreases at 9WD and three at 11WD.

The following populations were sorted:

- At 9WD: $GP2^+$ (triplicate), $GP2^-$ (triplicate), $E^{low}$ (duplicate) and M (triplicate)
- At 11WD: $GP2^{hi}$ (triplicate), $GP2^+$ (triplicate), $GP2^-$ (duplicate), $E^{low}$ (triplicate) and M (triplicate)

The third replicate for $E^{low}$ population at 9WD and $GP2^-$ population at 11WD were excluded due to the poor quality of the RNA.

