## [Decision Letter]

Thank you for submitting your article "Reconstructing human pancreatic organogenesis by mapping specific cell populations during development" for consideration by *eLife*. Your article has been favorably evaluated by Mark McCarthy (Senior Editor) and three reviewers, one of whom is a member of our Board of Reviewing Editors. The following individuals involved in review of your submission have agreed to reveal their identity: Susana Chuva de Sousa Lopes (Reviewer #2); Timo Otonkoski (Reviewer #3).

The reviewers have discussed the reviews with one another and the Reviewing Editor has drafted this decision to help you prepare a revised submission.

Summary:

The authors have undertaken a cellular deconstruction of how the human pancreas develops by studying the expression dynamics of several surface markers between 7 and 12 weeks of development. There is particular focus on the insulin-producing β cells. The evolution of these cell populations was very dynamic during the developmental stages studied, demonstrating that endocrine cells arise in the population that down-regulates both ECAD and GP2 expression. Further refinement of the characterization was achieved by including the cell surface marker SUSD2. Finally, human pluripotent stem cells undergoing pancreatic differentiation were analysed by the same parameters, demonstrating that in vitro development followed the same principles identified in vivo.

Essential revisions:

The three referee reports are included as all raise important points that should be addressed (and would improve the manuscript). The major request focuses on more detailed analysis and reporting of existing data (and not the acquisition of new 'wet lab' data). While all individual points should to be addressed in a revised manuscript and / or the accompanying rebuttal letter, there are some common themes to the referee reports:

1) There is a need to improve the detail of the computational and statistical analyses. There would be benefit from incorporating existing datasets on other pancreatic cell-types into some of the analyses (as highlighted by reviewer 2). This would then assist in commenting on other endocrine cell-types (as requested by reviewer 3) and examining other important genes (PAX4 is mentioned by reviewer 1).

2) There is scope to include more information on specific genes, for instance underlying the GO terms, either as part of the figures or as additional supplementary tables.

3) All the data should be made freely available for download.

4) The Materials and methods need far more detail to enable others to follow the work.

*Reviewer #1:*

Thank you very much for letting me review this manuscript. The major novelty is in the cellular deconstruction of how a human organ (the pancreas) develops. This has not been undertaken before and is challenging for a solid organ (e.g. compared to the haematopoietic system). The data have direct translational benefit for stem cell researchers. I make comments to try and improve the manuscript.

1) Figure 1. The data imply that a considerable number, 45% if I interpret the data correctly in Figure 1, of EpCAM+ cells either lack or (more likely) have lower levels of PDX1 and NKX6.1. These are then excluded from downstream analyses. The immunohistochemistry and text imply that all EpCAM+ cells possess PDX1, i.e. these cells are pancreatic progenitors. What do the authors think these cells EpCAM+ / low PDX1/low NKX6.1 cells are? How do these cells fit with the notion of NKX6.1 'low' cells as described by the authors in their recent Cell Reports paper, Ameri et al.? In that study on differentiating hPSCs, these cells were argued to be earlier stage pancreatic progenitors with higher levels of cell proliferation. Perhaps the authors could integrate their thoughts on this topic into the current manuscript?

2) At several places in the manuscript, there is mention of very precise timing, e.g. '7.4 WD'. What does this mean? How was such accuracy determined over and above 7WD or 8WD? Moreover, the authors described triplicate experiments: are these biological replicates at this very precise time point? Or technical replicates from a single human embryonic pancreas. The manuscript would benefit on full detail regarding human tissue use, perhaps as a supplementary table.

3) Computational analyses. A number of improvements should be made to allow others to gain maximum benefit from the data.a) For instance, in the subsection “GP2 and ECAD define 4 populations in the human fetal pancreatic epithelium that develop sequentially” (and elsewhere) (unless I missed it) it is not clear what cut-offs were applied to define particular genes as enriched. I don't think this detail was in the Materials and methods. b) The genes underlying all the GO terms should be listed, perhaps as a supplementary table as this would help the reader interpret the data. c) 'We correlated these signatures with RNAseq Single Cell data'. How were these correlations undertaken? d) The heatmaps in Figure 4 refer to very few genes. Why? And if selected, why were these particular genes chosen and (presumably) others overlooked?

4) Subsection “Acinar and endocrine functions segregate within the GP2 and ECAD populations”, end of first paragraph and elsewhere (e.g. subsection “CD142 and SUSD2 reveal heterogeneity within the GP2^-^ and E^low^ populations during development”, last paragraph): I would encourage the authors to avoid 'data not shown'. Either include the data or remove the interpretation.

5) I did not follow the text in the last paragraph of the subsection “CD142 and SUSD2 reveal heterogeneity within the GP2^-^ and E^low^ populations during development” referring to Figure 2—figure supplement 2. This figure supplement does not detail SUSD2 or CD142?

6) The broad transcriptomics data are useful but at present only a limited number of key genes are discussed. Would it be possible to describe some additional key factors, some of which have been notoriously difficult to track down accurately in native human embryonic/fetal pancreas, such as PAX4? The precision with which the authors have picked apart human pancreatic differentiation should offer an opportunity to narrow down when PAX4 becomes expressed and in which precise cell-type.

7) Figure 6. In the subsection “Endocrine progenitors develop in the GP2^-^ CD142^-^ SUSD2^-^ subset and mature within the E^low^SUSD2^+^ subset” the authors describe two different cell populations in which insulin is expressed in the earlier fetal pancreas at 8.6WD: an E^low^/CD142^-^/SUSD2^+^ population but not in the E^low^/CD142^-^/SUSD2^-^ population; and then the converse in older fetal pancreas at 10-12WD, namely in the E^low^/CD142^-^/SUSD2^-^ population but now not in the E^low^/CD142^-^/SUSD2^+^ population. At least to me, this is surprising as β cell differentiation operates over a window, i.e. β-cells are serially differentiating over time. Morphologically, there are differences-at the earlier time point β-cells tend to be scattered whereas at the later time point β-cells are more clustered. Therefore, my question is whether the authors have unearthed two distinct populations of β-cells?

*Reviewer #2:*

The authors have studied the expression dynamics of several surface markers in human pancreas from 7-12 weeks of development, mainly by FACS; and suggest an order of events during the differentiation of several lineages in the human pancreas.

1) The authors used single cell transcriptomics data from Segerstolpe et al. to generate Figure 3—figure supplement 1, but I don't understand what values were used to generate the heatmap. I assume they used RPKM values, but the legend goes from -1 to 1, so I am confused. I suggest that figure is replaced by heatmaps for the selected genes showing all individual cells instead of using averages (?) and using RPKM values.

2) For robustness, the set of cells from Muraro et al., 2016, Cell Systems, 3:385 should also be included in two extra independent heatmaps in Figure 3—figure supplement 1. The authors should also include not only acinar, ductal, α and β, but also the mesenchymal, δ, ε and pp cells.

3) The authors should use more than one HKG to normalize the QPCR results, the probes used are stable they could use bactin, HPRT and PPIA instead of just one of those (they already have the 3 probes in house).

4) The authors differentiate hPSCs into pancreas progenitors, but from the results it is unclear which of the 3 different lines mentioned were used and for what experiments. Can this be clarified in the Results and figures? Were the 3 lines used for the same experiments? And are the results comparable? In Figure 7: Could you specify what hPSCs were used for what experiments?

5) Discussion: you mention that culture of pancreatic cells have failed: could you be more specific about the reason for failure (cells don't attach; cells die; which conditions have you tried? What are the culture periods tried, etc.)?

6) The authors mention in the Discussion Ptf1, Cpa1 and cMyc as marking multipotent progenitors in mice. These genes were not analysed in the human data set. Were they expressed (and if yes, in what cells)? Could you include those genes in the heatmap in Figure 4?

7) I don't understand what the samples with "statistical significance" in all QPCR graphs were compared to? Could you clarify that in the figures?

8) Figure 2: the authors show the cells in the blue square stained with GP2 and ECAD. Could the authors also present the other 2 populations (in the two black squares) for the markers GP2 and ECAD? You should have that data acquired.

9) Figure 3: If the authors want to present PCA in 3D, they have to provide lines projecting each dot into the lower area. The plots as presented are 2D…

*Reviewer #3:*

In this study Ramond et al. have aimed to understand the sequential development of epithelial cell populations in the human fetal pancreas, focusing particularly on the emergence of the insulin-producing β cells. For this purpose, they have identified cell surface markers that were used to sort the cell populations between developmental stages estimated to represent 7-12 weeks. They identify a CD45-CD31-EPCAM+ population that includes the pancreatic progenitors. This population was then found to contain 4 subpopulations based on the expression of GP2 and E-Cadherin. The evolution of these cell populations was very dynamic during the developmental stages studied, demonstrating that endocrine cells arise in the population that downregulates both ECAD and GP2 expression. Further refinement of the characterization was achieved by including the cell surface marker SUSD2. Finally, human pluripotent stem cells undergoing pancreatic differentiation were analysed by the same parameters, demonstrating that the in vitro development follows the same principles identified in vivo.

Overall, the study represents an impressive amount of flow cytometry analysis in human fetal pancreas, taking into account the scarcity of this research material. These results contribute to the understanding of human pancreatic development and provide surface markers that could be utilized to isolate bona fide pancreatic endocrine progenitors. The novelty of the study relies in the combined used of previously reported and novel surface marker to delimit the target cell population, both in human fetal pancreas samples and stem cell derived cells. However, the novelty is decreased by the recent report by the same investigators (Ameri et al., Cell Reports 2017) in which GP2 was characterized as a surface marker for pancreatic endocrine progenitors in the human fetal pancreas and hPSCs.

The value of the study would be increased by adding the following analyses:

1) Out of all endocrine cells, the analysis focuses only on β cells. In which populations do the other major endocrine cell populations reside? It would be interesting to see at which stage and in which population within the GP2^-^ ECAD^low^ SUSD2^+^ are other endocrine cell hormones than insulin expressed: is the GCG^+^ population in SUSD2^+^ or SUSD2^-^? What is the distribution of hormone+ endocrine cells in the single cell samples (6D and 6E)? Are there polyhormonal cells, as is often seen in hPSC differentiation?

2) In the fourth paragraph of the Discussion the authors describe the GP2^+^ cell population to represent the multipotent "tip" cells described in the mouse embryo to express PTF1a, Cpa1 and c-Myc. Are these markers expressed in the human GP2^+^ cells?

The quality of the presentation and the statistical analysis should be improved:

1) Many of the figures are difficult to interpret. In general, it would be advisable to present as much as possible of the results as the quantitative summary of all experiments (as in Figure 2), and also include statistical comparison of the changes.

2) The immunofluorescent image in Figure 1 should be made larger and clearer.

3) The methods in general are only superficially described and there are no detailed supplementary methods.

---

## [Author Response]

*Reviewer #1:*

*[…] 1) Figure 1. The data imply that a considerable number, 45% if I interpret the data correctly in Figure 1, of EpCAM+ cells either lack or (more likely) have lower levels of PDX1 and NKX6.1. These are then excluded from downstream analyses. The immunohistochemistry and text imply that all EpCAM+ cells possess PDX1, i.e. these cells are pancreatic progenitors. What do the authors think these cells EpCAM+ / low PDX1/low NKX6.1 cells are? How do these cells fit with the notion of NKX6.1 'low' cells as described by the authors in their recent Cell Reports paper, Ameri et al.? In that study on differentiating hPSCs, these cells were argued to be earlier stage pancreatic progenitors with higher levels of cell proliferation. Perhaps the authors could integrate their thoughts on this topic into the current manuscript?*

The reviewer interpretation is correct. On Figure 1, 45% of the EPCAM+ cells express low levels of PDX1 and NKX6-1. In this context, it would really be interesting to compare the proliferative potential of EPCAM+ cells in relation to their NKX6-1 level. Indeed, defining cells that can be amplified from the human fetal pancreas is an important topic in the field. As suggested, we highlighted in the third paragraph of the Discussion section, the comparison with the Ameri et al. work of the new version of the manuscript.

*2) At several places in the manuscript, there is mention of very precise timing, e.g. '7.4 WD'. What does this mean? How was such accuracy determined over and above 7WD or 8WD?*

We determined the developmental age based on the length of the foot and the anatomy of the foot and the hand. With such information, the precise age was calculated using the formula from Manjunatha et al., 2012 (the Egyptian Journal of Forensic Sciences): Age = 4.11 X Foot Length + 5.60. We have now included this information in the Appendix subsection “Determination of the developmental stage”.

*Moreover, the authors described triplicate experiments: are these biological replicates at this very precise time point? Or technical replicates from a single human embryonic pancreas. The manuscript would benefit on full detail regarding human tissue use, perhaps as a supplementary table.*

All experiments represent *biological* replicates. We now added in the Appendix subsection “Replicates” that all replicates in the study are biological replicates.

*3) Computational analyses. A number of improvements should be made to allow others to gain maximum benefit from the data.a) For instance, in the subsection “GP2 and ECAD define 4 populations in the human fetal pancreatic epithelium that develop sequentially” (and elsewhere) (unless I missed it) it is not clear what cut-offs were applied to define particular genes as enriched. I don't think this detail was in the Materials and methods.*

We now describe this point in the Materials and methods subsection “Transcriptomic statistical analysis”: the cut-offs we applied to define “specifically enriched genes”: “Each population (GP2^hi^, GP2^+^, GP2^-^, E^low^ and M) was compared to the other populations using two-tailed Student t-test. Genes were considered as enriched in a specific population when over-expressed (p-value < 0,05 and Fold Change > 2) in a population compared to any of the others. The list of the “specific enriched genes” is displayed in Table I and II.”

*b) The genes underlying all the GO terms should be listed, perhaps as a supplementary table as this would help the reader interpret the data.*

We now added [Supplementary-material SD3-data] to describe the genes underlying all the GO terms.

*c) 'We correlated these signatures with RNAseq Single Cell data'. How were these correlations undertaken?*

[Supplementary-material SD3-data] displayed the lists of genes specifically enriched in GP2^hi^ and E^low^ populations. We next wanted to determine their expression pattern in the different adult human pancreatic cell types. For this purpose, we searched for the expression of these enriched genes in the Segerstolpe et al. 2016 single cell RNAseq data and generated heatmaps representing expression levels of the fetal enriched genes in the different adult pancreatic cell subpopulations. This is now described in the subsection “Transcriptomic statistical analysis”.

*d) The heatmaps in Figure 4 refer to very few genes. Why? And if selected, why were these particular genes chosen and (presumably) others overlooked?*

For Figure 4, we *selected* a limited number of genes known to be linked, based on the scientific literature, to pancreatic acinar and endocrine cell development. The term “selected” is now added in the second paragraph of the subsection “Acinar and endocrine functions segregate within the GP2 and ECAD populations”. We have now also added a new supplementary figure (Figure 3—figure supplement 2) based on gene ontology lists.

*4) Subsection “Acinar and endocrine functions segregate within the GP2 and ECAD populations”, end of first paragraph and elsewhere (e.g. subsection “CD142 and SUSD2 reveal heterogeneity within the GP2^-^ and E^low^ populations during development”, last paragraph): I would encourage the authors to avoid 'data not shown'. Either include the data or remove the interpretation.*

In the previous version of the manuscript, the term “data not shown” was used twice. It concerned:

1) the comparative analysis between the Segerstolpe et al. 2016 RNAseq and the list of the specific enriched genes in the E^low^ population at 9WD. We now deleted this information.

2) The expression of SUSD2 and CD142 at 8.6WD. Data are now shown in Figure 5—figure supplement 1.

*5) I did not follow the text in the last paragraph of the subsection “CD142 and SUSD2 reveal heterogeneity within the GP2^-^ and E^low^ populations during development” referring to Figure 2—figure supplement 2. This figure supplement does not detail SUSD2 or CD142?*

The reviewer is correct. Figure 2—figure supplement 2 detailed the decreased expression of ECAD in GP2^-^ population between 8.4 to 9.4WD. We now added Figure 5—figure supplement 1 in the new version of the manuscript that displays CD142 and SUSD2 expression at 8.6WD. We also modified the text in the subsection “CD142 and SUSD2 reveal heterogeneity within the GP2^-^ and E^low^ populations during development”.

*6) The broad transcriptomics data are useful but at present only a limited number of key genes are discussed. Would it be possible to describe some additional key factors, some of which have been notoriously difficult to track down accurately in native human embryonic/fetal pancreas, such as PAX4? The precision with which the authors have picked apart human pancreatic differentiation should offer an opportunity to narrow down when PAX4 becomes expressed and in which precise cell-type.*

In the new version of the manuscript, we now display two more heatmaps for the acinar and endocrine pathways based on the GSEA analysis in Figure 3—figure supplement 2.

Concerning PAX4: It is true that while the role of PAX4 in rodent β cell development has been demonstrated, its expression pattern and role in human remains unknown. We cannot exclude differences between rodent and human for PAX4 expression, as is the case for the transcription factor NKX2-2, a target of NEUROG3 in human (Jennings Diabetes 2013). From Figure 4, we have learned that *PAX4* expression is enriched in the E^low^ population both at 9 and 11WD. This information is indicated in the second paragraph of the subsection “Acinar and endocrine functions segregate within the GP2 and ECAD populations”. As a note for the reviewer, we have started to collect preliminary data that suggest that within the E^low^ population, PAX4 is first expressed by the E^low^SUSD2^+^ subset. In addition, in this population, PAX4 appears to be co-expressed with *NEUROG3*. We are also investigating if *INS* is also co-expressed with PAX4. As this set of information is preliminary at this stage, we decided not to include it in the new version of the manuscript.

*7) Figure 6. In the subsection “Endocrine progenitors develop in the GP2^-^ CD142^-^ SUSD2^-^ subset and mature within the E^low^SUSD2^+^ subset” the authors describe two different cell populations in which insulin is expressed in the earlier fetal pancreas at 8.6WD: an E^low^/CD142^-^/SUSD2^+^ population but not in the E^low^/CD142^-^/SUSD2^-^ population; and then the converse in older fetal pancreas at 10-12WD, namely in the E^low^/CD142^-^/SUSD2^-^ population but now not in the E^low^/CD142^-^/SUSD2^+^ population. At least to me, this is surprising as β cell differentiation operates over a window, i.e. β-cells are serially differentiating over time. Morphologically, there are differences-at the earlier time point β-cells tend to be scattered whereas at the later time point β-cells are more clustered. Therefore, my question is whether the authors have unearthed two distinct populations of β-cells?*

This is a very interesting question. At this point, we do not have proof, but want to test the hypothesis that E^low^/CD142^-^/SUSD2^+^*INS^+^* cells could represent transient progenitors or pre-β cells. For this purpose, we are currently assaying the differences between E^low^/CD142^-^/SUSD2^+^*INS^+^* and E^low^/CD142^-^/SUSD2^-^*INS^+^* cells by single cell qPCR. A working hypothesis is that E^low^/CD142^-^/SUSD2^+^*INS^+^* would be poly-hormonal, which would not be the case for the E^low^/CD142^-^/SUSD2^-^*INS^+^* cells. Again, performing this type of experiment takes time due to the limitation in the number of fetal pancreases we have access to. We now discuss this point in the fifth paragraph of the Discussion.

*Reviewer #2:*

*The authors have studied the expression dynamics of several surface markers in human pancreas from 7-12 weeks of development, mainly by FACS; and suggest an order of events during the differentiation of several lineages in the human pancreas.*

*1) The authors used single cell transcriptomics data from Segerstolpe et al. to generate Figure 3—figure supplement 1, but I don't understand what values were used to generate the heatmap. I assume they used RPKM values, but the legend goes from -1 to 1, so I am confused. I suggest that figure is replaced by heatmaps for the selected genes showing all individual cells instead of using averages (?) and using RPKM values.*

As proposed by the reviewer we deleted the previous Figure 3—figure supplement 1 and replaced it by new heat maps figure entitled Figure 3—figure supplement 3. Values are now presented in RPKM (log2), displaying all individual cells.

*2) For robustness, the set of cells from Muraro et al., 2016, Cell Systems, 3:385 should also be included in two extra independent heatmaps in Figure 3—figure supplement 1.*

We agree it would be interesting to include the data from Muraro et al., 2016. However the cut off used to define specific cell types (Β cell for example) is not given in the Muraro et al. publication. It is thus difficult to draw a heat map similar to the one we generated from the Segerstolpe et al. publication.

*The authors should also include not only acinar, ductal, α and β, but also the mesenchymal, δ, ε and pp cells.*

We agree and we are now displaying acinar, α, β, δ, epsilon, γ and ductal cells in the new Figure 3—figure supplement 3.

*3) The authors should use more than one HKG to normalize the QPCR results, the probes used are stable they could use bactin, HPRT and PPIA instead of just one of those (they already have the 3 probes in house).*

All qPCRs on the human fetal pancreas were assayed using *PPIA*. We now added this point in the Materials and methods.

We have compared *bactin, HPRT* and *PPIA* as house-keeping genes in the fetal human pancreas. *Bactin* was not selected as a house-keeping gene for the human fetal pancreas because it is expressed at different levels in the different cell populations. The use of *HPRT* was not convenient for our experiments where we are limited in the quantity of cDNA we are using for each qPCR reaction as it is detected at later cycles than PPIA.

*4) The authors differentiate hPSCs into pancreas progenitors, but from the results it is unclear which of the 3 different lines mentioned were used and for what experiments. Can this be clarified in the Results and figures? Were the 3 lines used for the same experiments? And are the results comparable? In Figure 7: Could you specify what hPSCs were used for what experiments?*

For Figure 7, we used AD3.1 iPSC. For Figure 7 we used 3 different lines: SA121 hESC, AD2.1 iPSC, AD3.1 iPSC. This information now appears in the legend of Figure 7, in the Results, and the Materials and methods subsection “Human pluripotent stem cells differentiation into pancreatic endocrine cells mimics human fetal endocrine cell development”. Results were comparable from one line to the other as displayed in Figure 7 by the limited variability observed from one experiment to the other.

*5) Discussion: you mention that culture of pancreatic cells have failed: could you be more specific about the reason for failure (cells don't attach; cells die; which conditions have you tried? What are the culture periods tried, etc.)?*

We faced 3 main issues. First we have been and remain limited by the number of fetal pancreases we have access to. Second, the number of sorted cells when using combinations of antibodies is low (between 1,000 and 5,000 cells per fraction) and culturing such a limited number of cells is really challenging. Third, during the culture period, half of the sorted cells die (even with ROCK inhibitor). Thus, at the end of the culture period, the remaining cells are extremely difficult to analyse. We clearly need more time and experiments to solve the problems we encounter. A possibility could be to use a feeder layer such as 3T3-J2 cells as recently described (Trott et al., Stem Cell Reports 8:1675–1688, 2017) to amplify early progenitors. We started to design such a type of approach.

*6) The authors mention in the Discussion Ptf1, Cpa1 and cMyc as marking multipotent progenitors in mice. These genes were not analysed in the human data set. Were they expressed (and if yes, in what cells)? Could you include those genes in the heatmap in Figure 4?*

According to our transcriptome data, *PTF1A, CPA1* and *MYC* are expressed in the human fetal pancreas and enriched in GP2^+^ compared to the E^low^ population. This information can be extracted from the GEO platform (GSE96697) where we uploaded our data. But we did not test them in qPCRs. Following comments from the three reviewers we removed this paragraph.

*7) I don't understand what the samples with "statistical significance" in all QPCR graphs were compared to? Could you clarify that in the figures?*

Thank you for pointing out this point. The comparisons are now directly highlighted on the figures. Please see Figure 4, Figure 6 and 7.

*8) Figure 2: the authors show the cells in the blue square stained with GP2 and ECAD. Could the authors also present the other 2 populations (in the two black squares) for the markers GP2 and ECAD? You should have that data acquired.*

We indeed had the data already acquired. Information on this topic now appears on Figure 2—figure supplement 1. The expression of GP2 and E-CADHERIN in CD45+/CD31+ is now presented in the red square and the expression of GP2 and E-CADHERIN in the CD45-CD31-EPCAM- in the green square.

*9) Figure 3: If the authors want to present PCA in 3D, they have to provide lines projecting each dot into the lower area. The plots as presented are 2D.*

We agreed and now display the PCA in 2D into 3 parts in Figure 3—figure supplement 1.

*Reviewer #3:*

*[…] The value of the study would be increased by adding the following analyses:*

*1) Out of all endocrine cells, the analysis focuses only on β cells. In which populations do the other major endocrine cell populations reside? It would be interesting to see at which stage and in which population within the GP2^-^ ECAD^low^ SUSD2^+^ are other endocrine cell hormones than insulin expressed: is the GCG^+^ population in SUSD2^+^ or SUSD2^-^? What is the distribution of hormone+ endocrine cells in the single cell samples (6D and 6E)? Are there polyhormonal cells, as is often seen in hPSC differentiation?*

Our data (Figure 4) indicate that *INS, GCG* and *GHRL* are expressed in the E^low^ population. This is the case both at 9 and 11WD. As pointed by the reviewer, in our previous single cell analysis (Figure 6), we did not include primers for hormones. We recently started to repeat the experiments looking at hormones following single cell qPCR. But, please keep into account that we remain limited by the scarcity of the fetal pancreases. The objective of the experiment is to further characterize the ECAD^low^ SUSD2^+^ and ECAD^low^ SUSD2^-^ subsets. We are for example asking whether poly-hormonal cells are detected. As a flavour, our first preliminary data indicate that ECAD^low^ SUSD2^-^ subset is highly enriched in *INS* mRNA when compared to ECAD^low^ SUSD2^+^ subset.

*2) In the fourth paragraph of the Discussion the authors describe the GP2^+^ cell population to represent the multipotent "tip" cells described in the mouse embryo to express PTF1a, Cpa1 and c-Myc. Are these markers expressed in the human GP2^+^ cells?*

According to our transcriptome data, *PTF1a*, *CPA1* and *cMYC* are expressed in the human fetal pancreas. And enriched in GP2^+^ compared to the E^low^ population. This information can be extracted from the GEO platform (GSE96697) where we uploaded our data. But we did not test them in qPCRs. Following comments from the three reviewers we removed this paragraph.

The quality of the presentation and the statistical analysis should be improved:

*1) Many of the figures are difficult to interpret. In general, it would be advisable to present as much as possible of the results as the quantitative summary of all experiments (as in Figure 2), and also include statistical comparison of the changes.*

Figure 2, Figure 5 and Figure 7 represent the quantitative summaries of the flow cytometry experiments with means ( ± SEM). We are now also providing the p values throughout the Results section.

*2) The immunofluorescent image in Figure 1 should be made larger and clearer.*

The previous image had been compressed and we are now providing a better quality image of Figure 1.

*3) The methods in general are only superficially described and there are no detailed supplementary methods.*

We now modified the Materials and methods section and added an Appendix— Methods supplementary information.

Specifically:

We added a paragraph to explain how we determined weeks of development (in Appendix—Methods supplementary information).

We recapitulated the number of biological replicates used in this study for each experiment (in Appendix—Methods supplementary information).

We reformatted the transcriptome part in the Materials and methods and detailed the cut off used for the transcriptome analysis.

We reformulated the way we compared our transcriptomic data with the single cell RNAseq from Segerstolpe et al. (in the Materials and methods).

We detailed the minor modifications of the protocol for the differentiation of the hPSC (in the Materials and methods).